# A *Drosophila* screen of schizophrenia genes highlights neural and glial MMPs in neuronal remodeling

Shir Keret[1,2], Hagar Meltzer[1,2], Neta Marmor-Kollet[1,2], Oren Schuldiner[1,2]

**Schizophrenia is a multifactorial neuropsychiatric disorder of complex and mostly unknown etiology, affected by genetic, developmental, and environmental factors. Neuroanatomical abnormalities, such as loss of gray matter, are apparent before the onset of symptoms, suggesting a neurodevelopmental origin. Indeed, it has been hypothesized, and recently experimentally supported, that schizophrenia is associated with dysregulation of developmental synaptic pruning. Here, we explore the molecular link between schizophrenia-associated genes and developmental neuronal remodeling. We focused on the *Drosophila* mushroom body, which undergoes stereotypic remodeling during metamorphosis. We conducted a loss-of-function screen in either glia or neurons of *Drosophila* homologs of human genes that are associated with schizophrenia based on genomic studies. Out of our "positive hits," we focused on matrix metalloproteinases. Our combinatorial loss-of-function experiments suggest that *Drosophila* matrix metalloproteinases are required both in neurons and in glia for the pruning of mushroom body axons. Our results shed new light on potential molecular players underlying neurodevelopmental defects in schizophrenia and highlight the advantage of genetically tractable model organisms in the study of human neurodevelopmental disorders.**

## Introduction

After its initial establishment, the developing nervous system of both vertebrates and invertebrates undergoes remodeling to shape its mature connectivity. In humans, developmental neuronal remodeling occurs postnatally and includes degenerative events such as retraction of synapses and large-scale axon elimination. Although these processes occur most prominently during the first 2 yr of life, they continue until after adolescence (Luo & O'Leary, 2005; Schuldiner & Yaron, 2015). Defects in neuronal remodeling have long been hypothesized to be one of the underlying causes of

neuropsychiatric disorders, including schizophrenia (SCZ; Feinberg, 1982; Penzes et al, 2011).

SCZ is a severe, multifactorial mental health condition affecting ~1% of the population. It is characterized by delusions, hallucinations, social withdrawal, and cognitive deficits, typically onsetting during the late teens to early adulthood (McCutcheon et al, 2020). It is known from MRI studies that structural abnormalities in the brain, such as reduction in gray matter and ventricular enlargement, may precede the onset of psychotic symptoms, implying neurodevelopmental origin (Suzuki et al, 2002; Brent et al, 2013; Zikidi et al, 2020; Omlor et al, 2025). It has been suggested, for many years, that the characteristic neuroanatomical alterations in individuals with SCZ are due to dysregulation of neuronal remodeling, specifically overpruning of synapses (Peter, 1979; Feinberg, 1982; Hoffman & Dobscha, 1989; Keshavan et al, 1994; Huttenlocher & Dabholkar, 1997). The first experimental evidence to support this hypothesis came several years ago, that SCZ is associated with the excessive expression of complement components leading to increased synapse elimination by microglia (Sekar et al, 2016). This work and others highlighted glia, including microglia and astrocytes, as key mediators of synapse pruning in normal development and in SCZ and other neuropathologies (reviewed in Laricchiuta et al [2024], Notter [2021], and Scott-Hewitt et al [2023]). Still, the molecular basis of SCZ risk and its link to abnormalities in neuronal remodeling remains mostly unknown.

*Drosophila melanogaster* is an ideal model to study neuronal remodeling, because of its unparalleled genetic toolkit combined with its massive stereotypic circuit remodeling during metamorphosis (Truman, 1990; Yaniv & Schuldiner, 2016). Our focus is on the *Drosophila* mushroom body (MB), a brain structure that functions as a center of olfactory learning and memory (Heisenberg, 1998). The MB is comprised of three types of intrinsic neurons, known as Kenyon cells (KCs), which are sequentially born from identical neuroblasts (Fig 1A). Out of the three KC types (γ, α′/β′, and α/β), only the first-born γ-KCs undergo stereotypic remodeling. When they initially grow, in the embryonic and early larval stages, γ-KCs project axons that bifurcate to form vertical and medial lobes. At

[1]Department of Molecular Cell Biology, Weizmann Institute of Science, Rehovot, Israel  [2]Department of Molecular Neuroscience, Weizmann Institute of Science, Rehovot, Israel

Correspondence: hagar.meltzer@weizmann.ac.il; oren.schuldiner@weizmann.ac.il
Neta Marmor-Kollet's present address is Department of Biology, Brandeis University, MA, USA

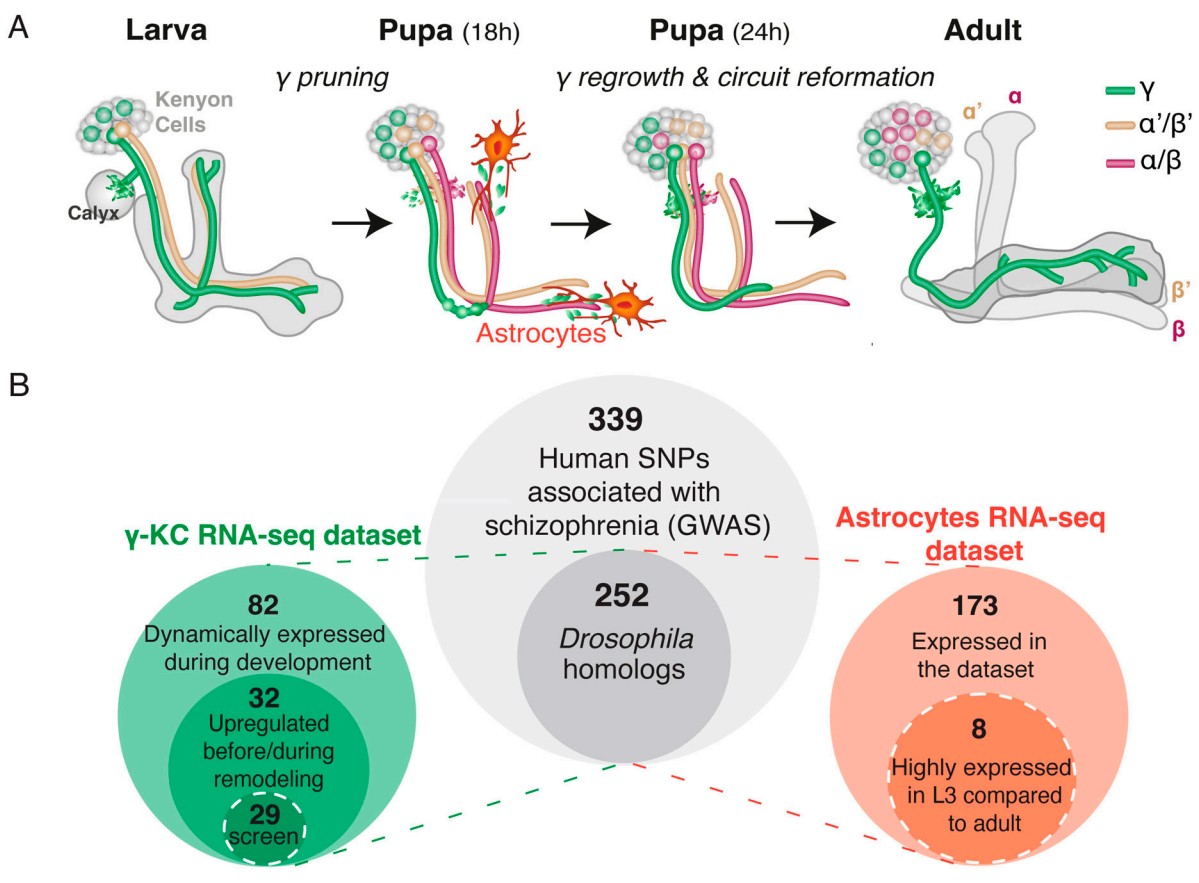

A

**Larva** **Pupa** (18h) **Pupa** (24h) **Adult**

*γ pruning* *γ regrowth & circuit reformation*

Kenyon Cells

Calyx

Astrocytes

γ
α'/β'
α/β

B

**339**
Human SNPs associated with schizophrenia (GWAS)

**252**
*Drosophila* homologs

**γ-KC RNA-seq dataset**

**82**
Dynamically expressed during development

**32**
Upregulated before/during remodeling

**29**
screen

**Astrocytes RNA-seq dataset**

**173**
Expressed in the dataset

**8**
Highly expressed in L3 compared to adult

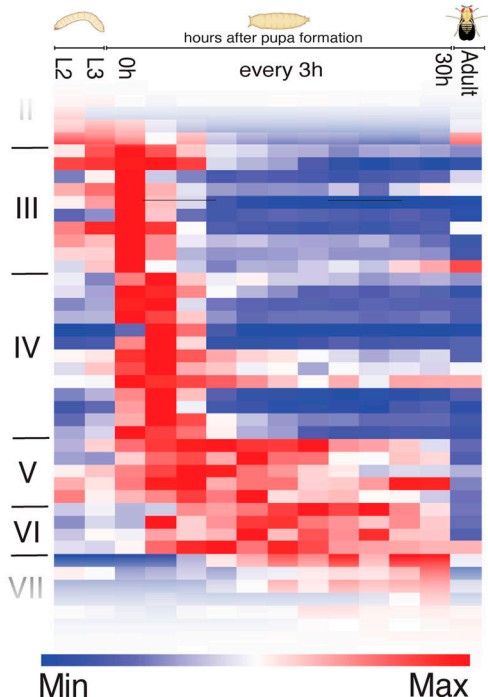

C Genes expressed in γ-KCs before/during remodeling

hours after pupa formation

L2 L3 0h every 3h 30h Adult

II
III
IV
V
VI
VII

Min Max

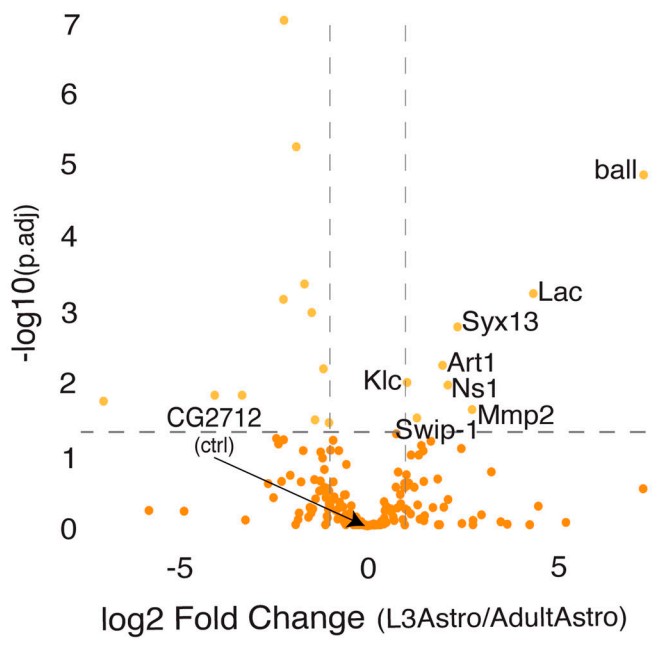

D Genes dynamically expressed in astrocytes

ball
Lac
Syx13
Art1
Klc
Ns1
Mmp2
CG2712 (ctrl)
Swip-1

-log10(p.adj)

log2 Fold Change (L3Astro/AdultAstro)

the early pupal stage (~6–18 h after puparium formation [h APF]) both axonal lobes prune up to their branchpoint. Subsequently, γ-KCs initiate axon regrowth to create the adult-specific medially projecting γ-lobe (Lee et al, 1999; Yaniv & Schuldiner, 2016; Fig 1A). Importantly, glia were shown, by our laboratory and others, to actively participate in the remodeling of γ-KCs (Awasaki & Ito, 2004; Watts et al, 2004; Awasaki et al, 2011; Hakim et al, 2014; Boulanger et al, 2021; Marmor-Kollet et al, 2023). Three glial subtypes are present around the MB: cortex glia, ensheathing glia, and astrocyte-like glia (hereafter referred to as astrocytes; Freeman, 2015). Astrocytes are the main scavengers of axonal debris after pruning (Hakim et al, 2014; Tasdemir-Yilmaz & Freeman, 2014). In addition, inhibiting astrocytic functions results in brain-wide defects in synapse elimination (Tasdemir-Yilmaz & Freeman, 2014). Moreover, we recently demonstrated that astrocytes actively infiltrate the axon bundle before pruning to facilitate axon defasciculation and elimination (Marmor-Kollet et al, 2023).

*Drosophila* is routinely used to study human neurodegenerative and neuropsychiatric disorders (Lessing & Bonini, 2009; van Alphen & van Swinderen, 2013), and SCZ-associated genes were shown to affect MB structure, activity, and function in memory and sleep (Sawamura et al, 2008; Furukubo-Tokunaga, 2009; Hidalgo et al, 2021). Here, we harness the fly genetic toolkit and the well-characterized remodeling of the MB to delve into the molecular underpinnings of the long-standing hypothesis that defects in neuronal remodeling contribute to SCZ etiology. We systematically examine how MB remodeling is affected by perturbations in *Drosophila* genes whose human homologs have been associated with SCZ. Our findings identify novel players in axon pruning and further highlight the significance of glial contribution to remodeling, providing a solid basis for future research on the neurodevelopmental molecular origin of SCZ.

# Results

## *Drosophila* homologs of human SCZ-associated genes are dynamically expressed in developing mushroom body γ-KCs and in astrocytes

We searched genome-wide association studies (GWAS) for candidate genes that have single nucleotide polymorphism (SNPs) in SCZ patients (339 genes; Hamshere et al, 2013; Wu et al, 2017; szdb.org/index.html), which is a compilation of data from the Psychiatric Genomics Consortium (PGC) and CLOZUK. We converted the SNP-containing genes to their *Drosophila* homologs using HumanMine (Lyne et al, 2022), resulting in 252 genes (Fig 1B; Table S1). To narrow

our candidate list to genes that are potentially required for MB pruning, we first used our previously generated transcriptional atlas of developing γ-KCs (Alyagor et al, 2018) to identify 196 genes that are expressed above threshold levels, out of which 82 showed dynamic expression (Fig S1). Next, we narrowed our list to genes that are specifically up-regulated during or before γ-axon pruning (Fig 1B and C, clusters III–VI highlighted). This process resulted in 32 candidate genes, out of which 29 had available RNA interference (RNAi) fly lines for loss-of-function (LOF) screening (Fig 1B; Table S1). Because of the known significance of glia in pruning, in parallel we also examined a second transcriptional dataset of larval and adult astrocytes, previously generated in the laboratory (Marmor-Kollet et al, 2023). Out of the 252 *Drosophila* homologs of SCZ-associated genes, 173 are expressed in astrocytes above threshold levels (Fig 1B; Table S1). Out of these, eight are up-regulated in late larval (third instar larvae, L3) compared with adult astrocytes (Fig 1D; Table S1), suggesting a potential role in axon pruning. All eight genes had available RNAi lines for screening.

## *Drosophila* homologs of human SCZ-associated genes are required in neurons and glia for γ-axon pruning

We screened candidate genes using a LOF strategy by tissue-specific expression of RNAi species via the Gal4-UAS system (Brand & Perrimon, 1993). For KD in γ-KCs, we used the pan-KC driver OK107-Gal4, which is strongly expressed in all KC types (α/β, α′/β′, and γ), combined with a second, independent binary system (Riabinina & Potter, 2016) to specifically visualize γ-KCs in the adult MB (R71G10-QF2 driving mtdT-HA; Fig 2A). Notably, although remodeling of MB γ-KCs occurs during metamorphosis, unpruned larval axons persist until adulthood. Thus, adult MBs reflect abnormalities that occurred during development. Out of the 29 genes we screened, OK107-Gal4–driven KD of 4 genes resulted in lethality, 11 genes displayed varying degrees of pruning defects ("positive hits"; Fig 2B–N), and one gene led to abnormal MB morphology (Fig S2A and B). In parallel, we used a similar strategy to screen the eight candidate genes in glia—by driving RNAi using the strong pan-glial driver Repo-Gal4, combined with R71G10-QF2–driven mtdT to visualize γ-KCs (Fig 3A). Strikingly, KD of seven of the eight genes resulted in γ-axon pruning defects (Fig 3B–J).

## *Drosophila* matrix metalloproteinases are required in KCs and glia for γ-axon pruning

Many of the "positive hits" in the screen are promising directions for further exploration. Interestingly, two genes—*syntaxin 13* (*Syx13*) and *matrix metalloproteinase 2* (*Mmp2*)—emerged as required for pruning

**Figure 1.** ***Drosophila* homologs of SCZ-associated genes are dynamically expressed in developing mushroom body γ-KCs and in astrocytes.**
**(A)** Schematic representation of *Drosophila* MB development. During early pupal stages, only the γ-KCs (green) prune their dendrites completely and their vertical and medial axonal branches up to their branchpoint. Subsequently, they reform dendrites in the calyx and regrow axons to form the adult-specific medial lobe. Later-born α′/β′- and α/β-KCs, as well as astrocytes, are also depicted. **(B)** Schematic representation describing the logic of candidate selection for the loss-of-function screen within KCs and glia. White dashed circles highlight the portion of genes that were screened. **(C)** Heatmap representing the expression of 32 genes (clusters III–VI) up-regulated during/after remodeling that were selected as candidates for screening (based on Alyagor et al [2018]). Each row represents a gene, with red and blue indicating high and low relative expression, respectively. The full heatmap of all dynamically expressed genes is available in Fig S1. **(D)** Volcano plot representing the comparison of gene expression between larval and adult astrocytes (based on Marmor-Kollet et al [2023]). Annotated genes are expressed significantly higher in L3 astrocytes compared with adult astrocytes. Each dot represents a gene, the x-axis represents the $\log_2$ fold change, and the y-axis represents the $-\log_{10}$ of the adjusted *P*-value (*P*.adj). For the full gene expression data and the candidate gene lists, see Table S1.

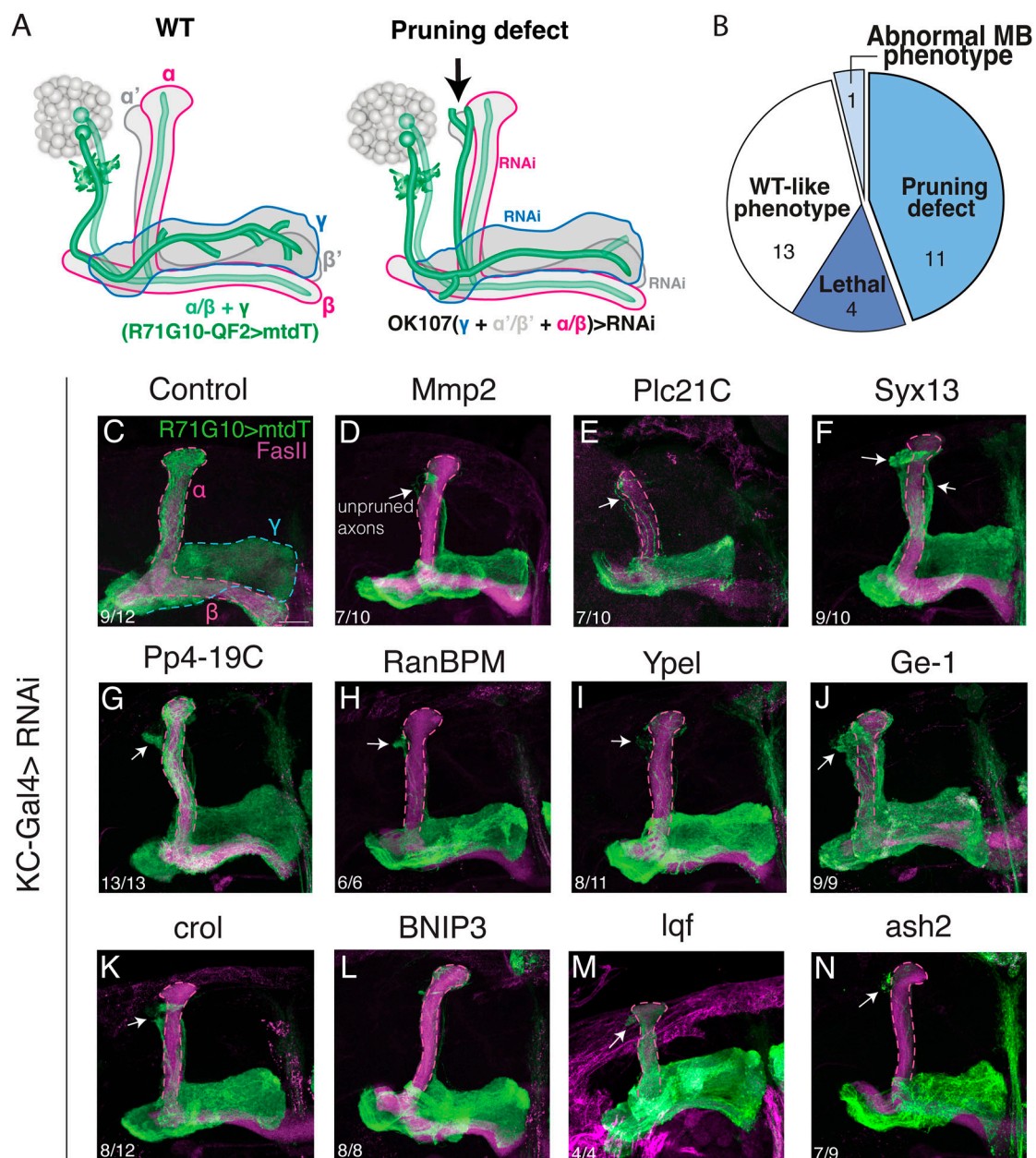

**Figure 2. *Drosophila* homologs of human SCZ-associated genes are required in neurons for γ-axon pruning.**

**(A)** Schematic representation of the experimental design. RNAi transgenes targeting candidate genes were driven in KCs by the strong driver OK107-Gal4 (γ—blue outline; α/β—magenta outline; α'/β'—gray). The γ-axon pruning phenotype was examined in adult brains via the expression of myristoylated tandem Tomato (mtdT; green) driven by R71G10-QF2. Left: WT, all γ-axons were pruned during pupal stages and regrew to form the adult-specific medial γ-lobe (blue outline). Note that R71G10-QF2 is also stochastically expressed in α/β neurons, which are in addition strongly positive to FasII (magenta outline). Right: pruning defect; some γ-axons were not pruned—remnants of larval projections (arrow) are visible outside the highly fasciculated α axon bundle (magenta outline). Adapted with permission from Marmor-Kollet et al (2023). **(B)** Pie chart representing the overall KD screen results in KCs (see Table S1; for the abnormal MB phenotype, see Fig S2). **(C, D, E, F, G, H, I, J, K, L, M, N)** Confocal z-projections of adult MBs in which the indicated UAS-RNAi transgenes were expressed in all KCs using OK107-Gal4, whereas R71G10-QF2 drives the expression of mtdT (green) in γ-KCs and stochastically in α/β-KCs. **(D, E, F, G, H, I, J, K, L, M, N)** RNAi transgenes target the following genes: *matrix metalloproteinase 2* (*Mmp2*; (D)), *phospholipase C at 21C* (*Plc21C*; (E)), *syntaxin 13* (*Syx13*; (F)), *protein phosphatase 19C* (*Pp4-19C*; (G)), *Ran-binding protein M* (*RanBPM*; (H)), *Yippee-like* (*Ypel*; (I)), *Ge-1* (*Ge-1*; (J)), *crooked legs* (*crol*; (K)), *BCL2 interacting protein 3* (*BNIP3*; (L)), *liquid facets* (*lqf*; (M)), *absent, small, or homeotic discs 2* (*ash2*; (N)). **(C)** Control is RNAi targeting luciferase (C). FasII antibody (magenta) strongly labels α/β-KCs and weakly labels γ-KCs. The γ-lobe is outlined in blue, and the α/β lobes are outlined in magenta. Arrows indicate unpruned axons. Scale bar corresponds to 20 μm. The number of MBs (each from an individual brain) showing the presented phenotype out of the total n for each genotype is indicated. For all screened gene genotype and phenotype findings, see Table S1. Lines used in the screen appear in Table 1. Full genotypes appear in Table 2.

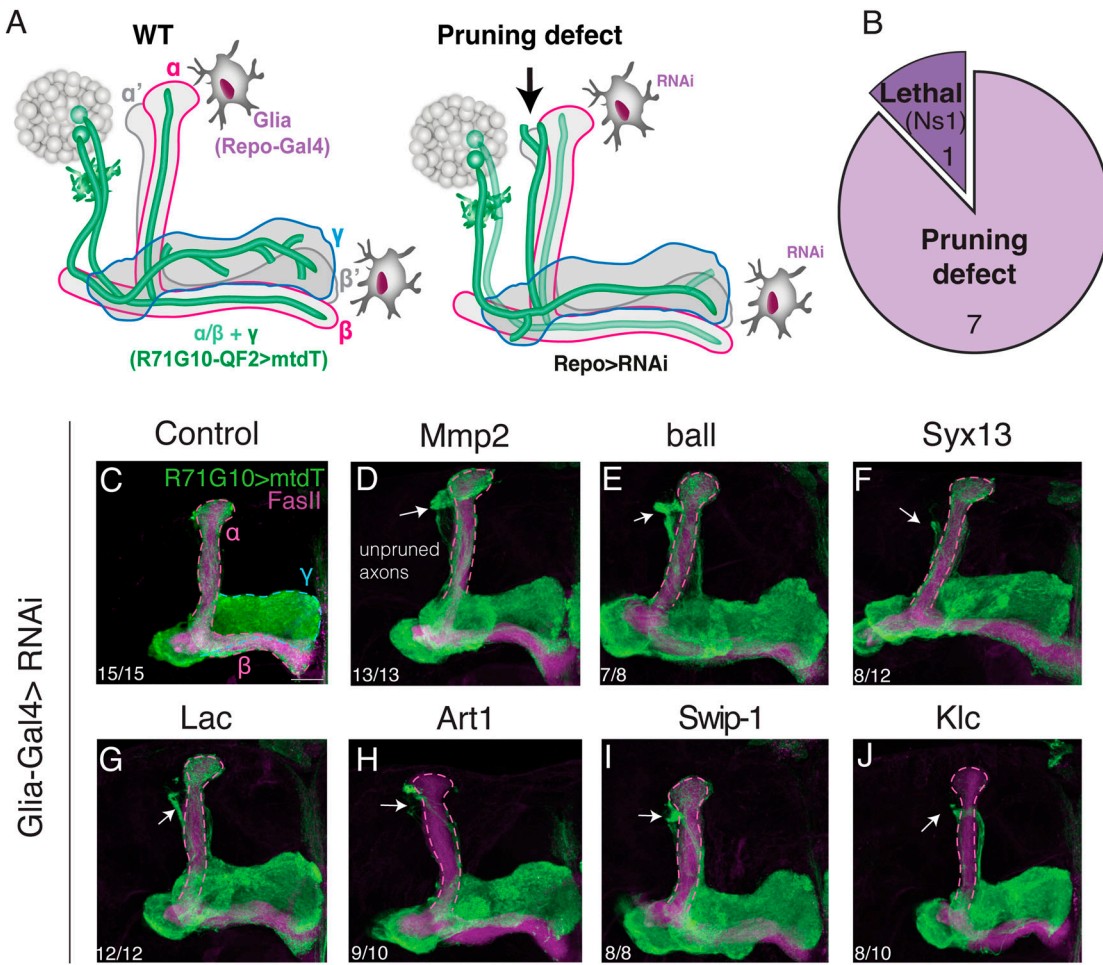

**Figure 3.** ***Drosophila* homologs of human SCZ-associated genes are required in glia for γ-axon pruning.**
**(A)** Schematic representation of the experimental design. RNAi transgenes targeting candidate genes were driven in all glia by Repo-Gal4 (gray cells). The γ-axon pruning phenotype was examined in adult brains via the expression of mtdT (green) driven by R71G10-QF2. **(B)** Pie chart representing the overall glial screen results (see Table S1). **(C, D, E, F, G, H, I, J)** Confocal z-projections of adult MBs in which the indicated UAS-RNAi transgenes were expressed in all glial cells using Repo-Gal4, whereas R71G10-QF2 drives the expression of mtdT (green) in γ-KCs and stochastically in α/β-KCs. **(D, E, F, G, H, I, J)** RNAi transgenes target the following genes: *matrix metalloproteinase 2* (*Mmp2*; (D)), *Ballchen* (*ball*; (E)), *syntaxin 13* (*Syx13*; (F)), *lachesin* (*Lac*; (G)), *arginine methyltransferase 1* (*Art1*; (H)), *swiprosin-1* (*Swip-1*; (I)), *kinesin light chain* (*Klc*; (J)). **(C)** Control is RNAi targeting *CG2712* (C), which is expressed in near-zero levels (in both larval and adult astrocytes), as indicated by its location at the origin of the volcano plot (see Fig 1D). FasII antibody (magenta) strongly labels α/β-axons and weakly labels γ-axons. The γ-lobe is outlined in blue, and the α/β lobes are outlined in magenta. Arrows indicate unpruned axons. Scale bar corresponds to 20 μm. The number of MBs (each from an individual brain) showing the presented phenotype out of the total n for each genotype is indicated. For all screened gene genotype and phenotype findings, see Table S1. Lines used in the screen appear in Table 1. Full genotypes appear in Table 2.

in both neurons and glia. We decided to delve deeper into *Mmp2*, as a previous, CRISPR-based screen done by our laboratory also found it to be required in γ-KCs for axon pruning (Meltzer et al, 2019).

MMPs are extracellular proteases that cleave ECM components (Page-McCaw et al, 2007; Sreesada et al, 2025). In the mouse, there are 23 *MMPs*, whereas the zebrafish genome encodes 25 *MMPs*. The *Drosophila* genome, in contrast, encodes only two *MMPs*—*Mmp1* and *Mmp2*—which together have 13 predicted isoforms, most of which are likely secreted, whereas each *Mmp* also has a glyco-sylphosphatidylinositol (GPI)-anchored isoform (LaFever et al, 2017). The significantly reduced complexity makes *Drosophila MMPs* an ideal model system to uncover novel insights into their roles in neuronal remodeling. Our transcriptional datasets indicate that both MMPs are dynamically expressed in γ-KCs and in larval astrocytes

(Fig 4A). We thus decided to KD *Mmp1* in both KCs and glia, and compare it with *Mmp2* (Fig 4B–I). Blinded ranking by two independent investigators (Fig S3) established that the pruning defect induced by KD of *Mmp1* or *Mmp2* is significant compared with controls in glia (Fig 4B–E). KD of *Mmp1* in KCs results in a mild pruning defect, not reaching statistical significance, whereas knockdown of *Mmp2* is significant compared with controls (Fig 4F–I). Importantly, we validated the efficiency of the *Mmp1* RNAi line in reducing protein levels using an Mmp1 antibody (Fig S4A–E). The specific *Mmp2* RNAi line that we used was previously demonstrated to efficiently reduce Mmp2 protein levels (Harmansa et al, 2023).

We next used CRISPR/Cas9 to generate a novel, predicted null *mmp2* mutant allele. Because of its (expected) lethality, we employed MARCM (Lee et al, 1999) to generate homozygous mutant γ-KC clones.

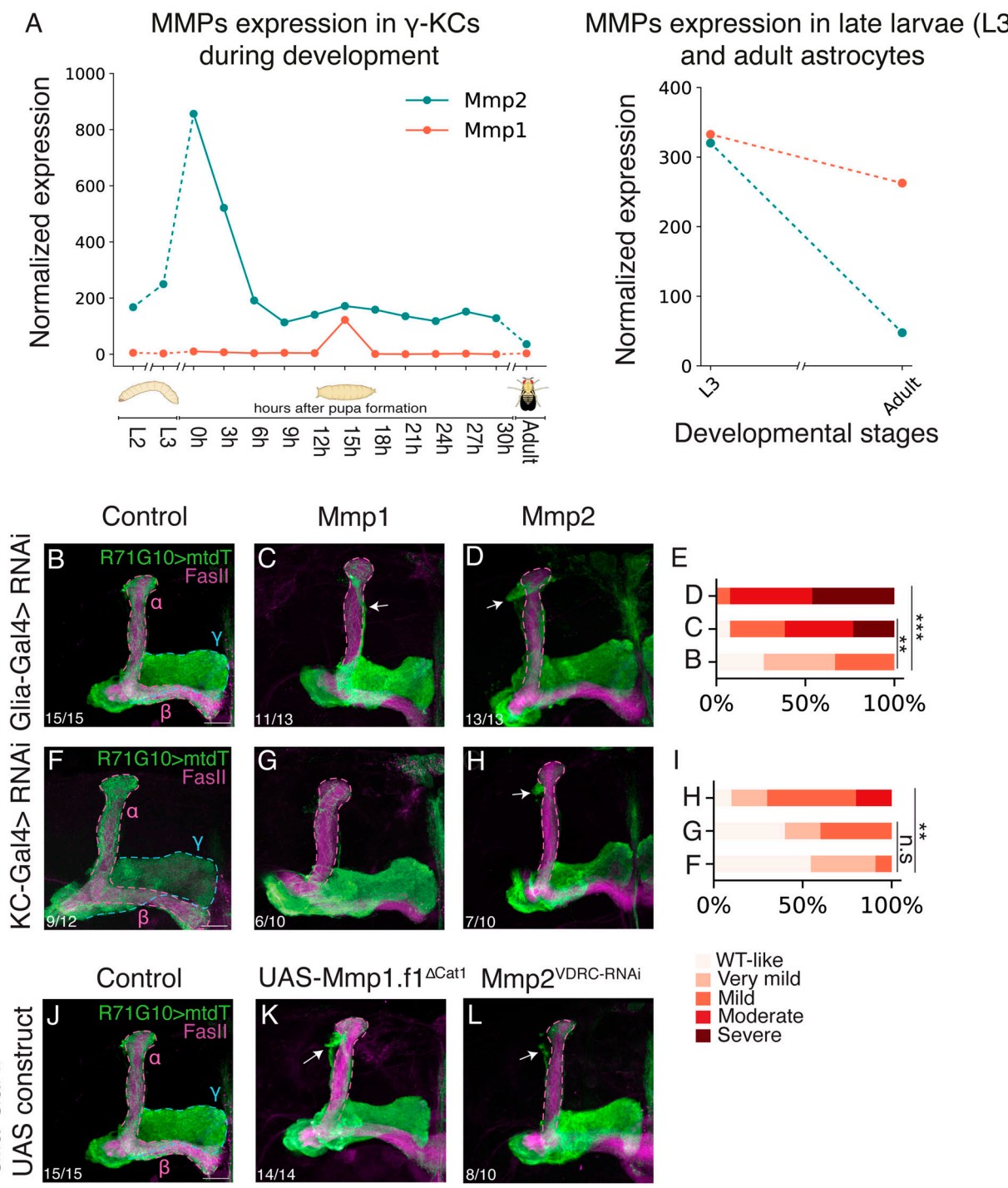

**Figure 4. *Drosophila* MMPs are required in KCs and glia for γ-axon pruning.**
**(A)** Normalized gene expression of *Mmp1* (orange) and *Mmp2* (cyan) in developing γ-KCs (left; based on Alyagor et al [2018]) and in larval/adult astrocytes (right; based on Marmor-Kollet et al [2023]). **(B, C, D, F, G, H)** Confocal z-projections of adult MBs in which γ-KCs are labeled by R71G10-QF2–driven QUAS-mtdT (green). **(B, C, D, F, G, H)** RNAi species targeting control genes (B, F), *Mmp1* (C, G), or *Mmp2* (D, H) are expressed in all glia using Repo-Gal4 (B–D), or in all KCs using OK107-Gal4 (F–H). Control in (B) is RNAi targeting *CG2712* (see Fig 1D), and in (F) is RNAi targeting luciferase (notably, 9 out of 12 MBs had WT-like phenotypes, whereas 3 out of 12 showed very mild pruning defects). **(E, I)** Quantification of the pruning defect phenotypes, ranging from 1 (WT-like phenotype) to 5 (severe pruning defect). Genotypes are indicated with the letter of the corresponding image within this figure. B versus C, *P* = 0.001; B versus D, *P* = 0.0001; F versus G, *P* = 0.289; F versus H, *P* = 0.009. ***P* < 0.001; *P* < 0.05; n.s., not significant. For examples of quantification scores, as well as ranking comparison, see Fig S3. **(J, K, L)** Confocal z-projections of adult MBs in which γ-KCs are labeled by R71G10-QF2–driven QUAS-mtdT (green). Indicated UAS transgenes are driven by Repo-Gal4. Control in (J) is RNAi targeting *CG2712* (see Fig 1D). Arrows indicate unpruned axons. The γ-lobe is outlined in blue and the α/β lobes in magenta. FasII antibody (magenta) strongly labels α/β-axons and weakly labels γ-axons. Scale bar corresponds to 20 μm.

However, $mmp2^{\Delta54–120.PC}$ clones did not present pruning defects (Fig S5A–C), consistent with its predicted function as a predominantly secreted protein, and therefore a non–cell-autonomous role. Thus, to further validate our findings, we tested a second RNAi line targeting *Mmp2*, which similarly showed pruning defects when driven in glia (Fig 4J and L), but not in KCs (Fig S6A and C).

To further explore the requirement of *Mmp1*, we tested a dominant negative (DN) variant previously generated by the Page-McCaw laboratory (Glasheen et al, 2009). When expressed in glia, Mmp1-DN resulted in a severe γ-axon pruning defect (Fig 4J and K), whereas expressing it in KCs results in a mild pruning defect (Fig S6A and B), reinforcing the reduced requirement of *Mmp1* in γ-KCs compared with glia.

Notably, MMPs in both vertebrates and invertebrates are inhibited by TIMPs (tissue inhibitor of metalloproteinases; Wen et al, 2020)—one in *Drosophila*, and four in vertebrates. We thus expected that the over-expression of *Drosophila Timp* would mimic the *MMP* KD phenotype. However, overexpressing *Timp* in glia did not affect pruning, and in KCs only induced a mild and inconsistent pruning defect (Fig S7A–D). The lack of phenotype may result from a technical issue such as low ex-pression or protein mislocalization, or a biological issue such as more complicated involvement of *Timp* or other *MMP* inhibitors.

Importantly, the initial growth of γ-KC axons, as evident at the onset of pupariation, is unaffected by KD of either *Mmp1* or *Mmp2*, in neither KCs nor glia (Fig S8A–F), highlighting their specific role in the axon remodeling phase.

Taken together, our data suggest that both MMPs are required in glia and in KCs for γ-axon remodeling.

### Analysis of combinatorial MMP knockdown in both KCs and glia

Because most Mmp1 and Mmp2 isoforms are secreted, we spec-ulated that both are secreted in parallel from KCs and glia to promote axon pruning. Therefore, we performed different single and double KD combinations of *Mmp1* and/or *Mmp2* in KCs and/or glia (Fig 5A–C, E–G, and I–K). Our data and quantification (by two independent investigators; see the Materials and Methods section and Fig S3) draw two main conclusions: first, although both *Mmp1* and *Mmp2* are required for pruning, using the available drivers and RNAi reagents, the observed phenotypes of *Mmp2* KD are more severe in all conditions (Fig 5D, H, and L). Second, al-though *Mmp1/2* expression from both KCs and glia is required for pruning of γ-axons, the main source is likely glia, as KD of a single or double *MMP* in both glia and neurons did not further exacerbate the pruning defect severity compared with the respective KD in glia only (Fig 5M–O).

Because of the major glial contribution, combined with the known astrocytic involvement in remodeling and the expression data (Fig 4A), we decided to also test KD using the astrocyte-specific driver alrm-Gal4, which indeed resulted in a mild pruning defect in the case of *Mmp2* (but not for *Mmp1*; Fig S9A–C). The mild phenotype can be explained by weaker Gal4 expression compared with Repo-Gal4, or due to involvement of additional glial subtypes. Unfortunately, our attempts to analyze other glial subtypes were abruptly terminated (see the Acknowledgements section).

Taken together, our results suggest that although both *MMPs* affect pruning likely via secretion from both neurons

and glia, the strongest requirement for MB γ-axon pruning comes from MMP secretion by glia—at least partially by astrocytes.

## Discussion

Four decades ago, Feinberg hypothesized that SCZ is caused by aberrant synaptic pruning during adolescence: "too many, too few, or the wrong synapses are eliminated" (Feinberg, 1982). In recent years, accumulating findings are providing experimental support of this hypothesis (Sekar et al, 2016; Sellgren et al, 2019; Keshavan et al, 2020). In this work, we aimed to further explore the molecular as-sociation of SCZ-related genes with defective neuronal remodeling, using the stereotypic remodeling of the *Drosophila* MB.

In a genetic LOF screen for *Drosophila* homologs of human genes that contain SNPs associated with SCZ, about 40% of the genes we screened within the neurons (11 out of 29) were found to have a role in γ-axon pruning. In addition, seven out of eight genes we screened within the glia were found to have a role in γ-axon pruning. This high rate of "positive hits" strengthens the link be-tween SCZ-associated genes and defects in developmental neu-ronal remodeling and emphasizes glia's major role. Notably, we focused on genes that are up-regulated before pruning, but future efforts to overexpress genes that are down-regulated before pruning might also reveal interesting insights.

Although many of the "positive hits" are fascinating candidates for further research, in this study we decided to focus on the role of *MMPs* in neuronal remodeling, for several reasons. First, *Mmp2* was found to be required in both glia and neurons for γ-axon pruning. This is of particular interest because of the speculated role of glia in abnormal synaptic pruning in SCZ patients. Second, previous reports have highlighted the link between *MMPs* and the patho-physiology of SCZ. Specifically, elevated levels of *MMPs*, most prominently *Mmp9*, were shown to correlate with SCZ risk and cognitive impairment (Kudo et al, 2020; Schoretsanitis et al, 2021; Seitz-Holland et al, 2022; Dickerson et al, 2023). Although it has been suggested that this correlation is via *MMP*-dependent changes in dendritic spine morphology (Lepeta et al, 2017), the precise mechanism remains poorly understood. Of note, the *Drosophila MMPs* are evolutionarily conserved (Llano et al, 2002; LaFever et al, 2017), highlighting their potential relevance and providing a simplified and powerful model system for mechanistic exploration.

*MMPs* are known for their roles in tissue remodeling by cleaving substrates in the ECM (Page-McCaw et al, 2007), and are well studied in the cancer field (Overall & López-Otín, 2002; Cox, 2021), but their mechanistic function in the developing nervous system is less characterized (e.g., Broadie et al, 2011; Dear et al, 2016; Gore et al, 2021). MMPs are known to be secreted by neurons and glia, and are thought to facilitate synaptic remodeling of dendritic spines (Huntley, 2012). In adult rodent brains, pharmacologically (Szklarczyk et al, 2002) or injury (Pijet et al, 2019)-induced dendritic remodeling is suggested to occur via *Mmp9*. In the developmental remodeling of *Drosophila* dendritic arborization (da) neurons, both *Mmp1* and *Mmp2* were shown to be required for the

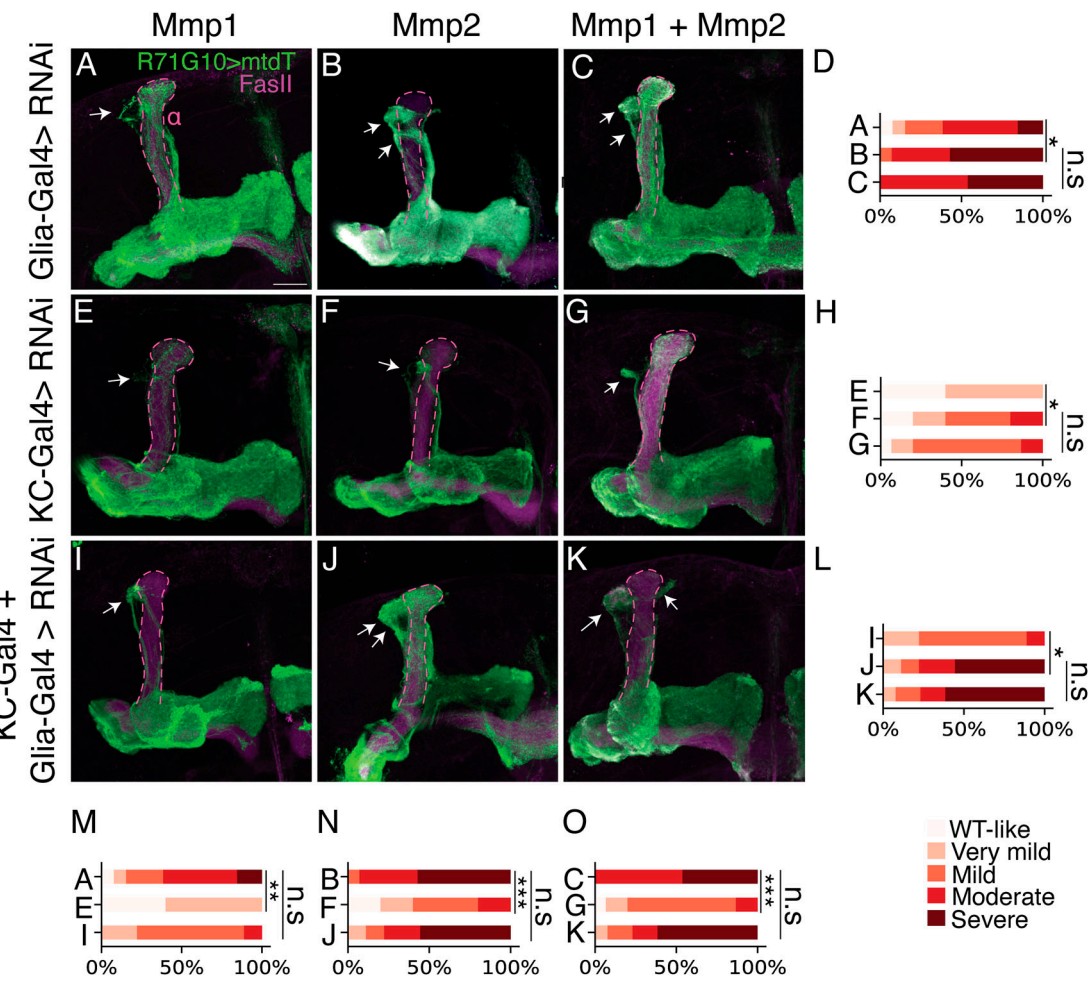

**Figure 5. Analysis of combined MMP knockdown in both KCs and glia.**
**(A, B, C, E, F, G, I, J, K)** Confocal z-projections of adult MBs in which γ-KCs are labeled by R71G10-QF2–driven QUAS-mtdT (green), and UAS-RNAi species targeting *Mmp1* (A, E, I), *Mmp2* (B, F J), or both (C, G, K) are driven by Repo-Gal4 (A, B, C), OK107-Gal4 (E, F, G), or both simultaneously (I, J, K). FasII antibody (magenta) strongly labels α/β-axons and weakly labels γ-axons. Arrows indicate unpruned axons. The α-lobe is outlined in magenta. Scale bar corresponds to 20 μm. **(D, H, L, M, N, O)** Quantification of the pruning defect phenotypes, ranging from 1 (WT-like phenotype) to 5 (severe pruning defect). Genotypes are indicated with the letter of the corresponding image within this figure. For quantification score examples and ranking comparison, see Fig S3. ***P < 0.001; *P < 0.05; n.s., not significant. A versus B, P = 0.023; B versus C, P = 0.860; E versus F, P = 0.046; F versus G, P = 0.719; I versus J, P = 0.023; J versus K, P = 0.943; A versus E, P = 0.001; A versus I, P = 0.092; B versus F, P = 0.0009; B versus J, P = 0.860; C versus G, P = 0.0003; C versus K, P = 0.943.

elimination of severed dendritic branches in a non–cell-autonomous manner (Kuo et al, 2005). Their source, as well as the mechanism by which they promote dendrite removal, remains unknown (the authors speculate they are secreted from phagocytic cells—at the time they suggested blood cells, but given more recent studies from the same laboratory, epidermal cells, are more likely; Han et al, 2014). Interestingly, *MMPs* were also shown to mediate dendrite reshaping of da neurons in the mature fly via local degeneration of the basement membrane, further reinforcing their significance in refining the mature nervous system (Yasunaga et al, 2010). Finally, it was shown that *Mmp1* is up-regulated in astrocytes after traumatic brain injury (Li et al, 2024), and in ensheathing glia after ventral nerve cord injury to promote axonal debris clearance (Purice et al, 2017).

The mechanism by which *MMPs* promote remodeling of MB γ-axons, and specifically their substrates in this context,

remains to be uncovered. Because the main known role of *MMPs* is cleaving ECM substrates, they may act as "path cleaners" for the migration of cells and molecules (Page-McCaw et al, 2007). This function can be relevant to axon pruning in several, nonmutually exclusive ways. One option is that *MMPs* allow the migration of myoglianin (Myo), a *Drosophila* TGF-β, in the ECM. *Myo* is part of the pruning initiation cascade and was shown to be secreted by cortex glia and astrocytes (Awasaki et al, 2011) to eventually promote the expression of *ecdysone receptor (EcR)-B1*, a key regulator of remodeling (Zheng et al, 2003; Yu et al, 2013). Moreover, it was shown that the human *Mmp9* can proteolytically activate latent TGF-β (Yu & Stamenkovic, 2000). However, we found that the expression of *EcR-B1* in γ-KCs is unaffected by glia or neuronal KD of *Mmp1/Mmp2* (Fig S10A–F). Another option is that *MMPs* facilitate astrocyte infiltration into the γ-axon bundle at the onset of pruning, which was shown to

**Table 1.   RNAi lines used in the KC screen (using OK107-Gal4) and the pan-glial screen (using Repo-Gal4) and their KD phenotypes.**

| Gene | TRiP# | BDSC# | Phenotype |
|---|---|---|---|
| KC screen | | | |
| Tartan | HM05011 | 28525 | WT-like |
| Fish-lips | HM05054 | 28568 | WT-like |
| Retinal degeneration A | JF03371 | 29435 | WT-like |
| Ppr-Y | HMS00819 | 33882 | WT-like |
| FER tyrosine kinase | HMS00249 | 33375 | WT-like |
| Matrix metalloproteinase 2 | HMJ23143 | 61309 | Pruning defect |
| Phospholipase C at 21C | HMS00436 | 32438 | Pruning defect |
| Proteasome alpha3 subunit | HMS05889 | 77145 | Lethal |
| Regulatory particle non-ATPase 7 | HMS00096 | 34787 | Lethal |
| Syntaxin 13 | HMS01723 | 38525 | Pruning defect |
| Anterior pharynx defective 1 | HMS01693 | 38249 | Lethal |
| BNIP3 | HMJ02058 | 42494 | Pruning defect |
| Liquid facets | HMJ22079 | 58130 | Mild pruning defect |
| Eukaryotic translation release factor 1 | HMS05812 | 67900 | Abnormal MB |
| DIP-kappa | HM04050 | 31740 | WT-like |
| MYPT-75D | HMS01166 | 34688 | WT-like |
| Protein phosphatase 19C | HMS01841 | 38372 | Pruning defect |
| GTPase-activating protein and centrosome-associated | HMS01132 | 34976 | WT-like |
| Ran-binding protein M | HMC05142 | 61172 | Pruning defect |
| Na[+]-driven anion exchanger 1 | HMC05184 | 62177 | WT-like |
| Minidiscs | HMC05214 | 62207 | WT-like |
| Serine-arginine protein kinase at 79D | HMC04154 | 55881 | WT-like |
| Ypel | HM05209 | 29530 | Pruning defect |
| Akt1 | HMS00007 | 33615 | Lethal |
| Absent, small, or homeotic discs 2 | HMC05816 | 64942 | Mild pruning defect |
| Fmr1 | HMS00248 | 34944 | WT-like |
| Arginine methyltransferase 1 | GL01072 | 36891 | WT-like |
| Ge-1 | HMS00340 | 32349 | Pruning defect |
| Crooked legs | HMS02202 | 41669 | Pruning defect |
| Pan-glial screen | | | |
| Ballchen | HMC04017 | 55330 | Pruning defect |
| Lachesin | HMS01756 | 38536 | Pruning defect |
| Syntaxin 13 | HMS01723 | 38536 | Pruning defect |
| Arginine methyltransferase 1 | JF01306 | 31348 | Pruning defect |
| Nucleostemin 1 | HMJ23534 | 61950 | Lethal |
| Matrix metalloproteinase 2 | HMJ23143 | 61309 | Pruning defect |
| Kinesin light chain | HMS02429 | 42597 | Pruning defect |
| Swiprosin-1 | JF01160 | 31585 | Pruning defect |

be required for defasciculation (Marmor-Kollet et al, 2023) and subsequent engulfment of the axonal debris (Hakim et al, 2014; Tasdemir-Yilmaz & Freeman, 2014). Unfortunately, testing this hypothesis requires three binary systems and is beyond the scope of this study. MMPs were also shown to degrade adhesion molecules (Page-McCaw et al, 2007), and our laboratory previously showed that the adhesion molecule fasciclin II (FasII) must be down-regulated for pruning to occur properly

**Table 2. Full *Drosophila* genotypes.**

| Figure panel | Full genotype | Comment |
|---|---|---|
| Fig 2C | y$^1$ v$^1$/+ or Y; R71G10-QF2, QUAS-mtdT:3xHA/+; P{y[+t7.7] v [+t1.8]=TRiP.JF01355}attP2/+; OK107-Gal4/+ | UAS-luciferase RNAi (BL#31603) |
| Fig 2D | y$^1$ v$^1$/+ or Y; R71G10-QF2, QUAS-mtdT:3xHA/P{y[+t7.7] v [+t1.8]=TRiP.HMJ23143}attP40; OK107-Gal4/+ | UAS-Mmp2-RNAi (BL#61309) |
| Fig 2E | y$^1$ sc* v$^1$ sev$^{21}$/+ or Y; R71G10-QF2, QUAS-mtdT:3xHA/+; P{y[+t7.7] v[+t1.8]=TRiP.HMS00436}attP2/+; OK107-Gal4/+ | UAS-Plc21c-RNAi (BL#32438) |
| Fig 2F | y$^1$ sc* v$^1$ sev$^{21}$/+ or Y; R71G10-QF2, QUAS-mtdT:3xHA/P{y [+t7.7] v[+t1.8]=TRiP.HMS01723}attP40; OK107-Gal4/+ | UAS-Syx13-RNAi (BL#38525) |
| Fig 2G | y$^1$ sc* v$^1$ sev$^{21}$/+ or Y; R71G10-QF2, QUAS-mtdT:3xHA/+; P{y [+t7.7] v[+t1.8]=TRiP.HMS01841}attP2/+; OK107-Gal4/+ | UAS-Pp4-19C-RNAi (BL#38372) |
| Fig 2H | y$^1$ sc* v$^1$ sev$^{21}$/+ or Y; R71G10-QF2, QUAS-mtdT:3xHA/P{y [+t7.7] v[+t1.8]=TRiP.HMC05142}attP40; OK107-Gal4/+ | UAS-RanBPM-RNAi (BL#61172) |
| Fig 2I | y$^1$ v$^1$/+ or Y; R71G10-QF2, QUAS-mtdT:3xHA/+; P{y[+t7.7] v [+t1.8]=TRiP.HM05209}attP2/+; OK107-Gal4/+ | UAS-Ypel-RNAi (BL#29530) |
| Fig 2J | y$^1$ sc* v$^1$ sev$^{21}$/+ or Y; R71G10-QF2, QUAS-mtdT:3xHA/+; P{y [+t7.7] v[+t1.8]=TRiP.HMS00340}attP2/+; OK107-Gal4/+ | UAS-Ge-1-RNAi (BL#32349) |
| Fig 2K | y$^1$ v$^1$/+ or Y; R71G10-QF2, QUAS-mtdT:3xHA/P{y[+t7.7] v [+t1.8]=TRiP.HMS02202}attP40; OK107-Gal4/+ | UAS-crol-RNAi (BL#41669) |
| Fig 2L | y$^1$ v$^1$/+ or Y; R71G10-QF2, QUAS-mtdT:3xHA/P{y[+t7.7] v [+t1.8]=TRiP.HMJ02058}attP40; OK107-Gal4/+ | UAS-BNIP3-RNAi (BL#42494) |
| Fig 2M | y$^1$ v$^1$/+ or Y; R71G10-QF2, QUAS-mtdT:3xHA/P{y[+t7.7] v [+t1.8]=TRiP.HMJ22079}attP40; OK107-Gal4/+ | UAS-lqf-RNAi (BL#58130) |
| Fig 2N | y$^1$ sc* v$^1$ sev$^{21}$/+ or Y; R71G10-QF2, QUAS-mtdT:3xHA/P{y [+t7.7] v[+t1.8]=TRiP.HMC05816}attP40; OK107-Gal4/+ | UAS-ash2-RNAi (BL#64942) |
| Fig S2A | y$^1$ v$^1$/+ or Y; R71G10-QF2, QUAS-mtdT:3xHA/+; P{y[+t7.7] v [+t1.8]=TRiP.JF01355}attP2/+; OK107-Gal4/+ | UAS-luciferase RNAi (BL#31603) |
| Fig S2B | y$^1$ v$^1$/+ or Y; R71G10-QF2, QUAS-mtdT:3xHA/P P{y[+t7.7] v [+t1.8]=TRiP.HMS05812}attP40; OK107-Gal4/+ | UAS-eRF1-RNAi (BL#67900) |
| Fig 3C | y$^1$ sc* v$^1$ sev$^{21}$/+ or Y; R71G10-QF2, QUAS-mtdT:3xHA/P{y [+t7.7] v[+t1.8]=TRiP.HMC04724}attP40; Repo-Gal4 UAS-CD8:: GFP/+ | UAS-CG2712-RNAi (BL#57418) |
| Fig 3D | y$^1$ v$^1$/+ or Y; R71G10-QF2, QUAS-mtdT:3xHA/P{y[+t7.7] v [+t1.8]=TRiP.HMJ23143}attP40; Repo-Gal4 UAS-CD8::GFP/+ | UAS-Mmp2-RNAi (BL#61309) |
| Fig 3E | y$^1$ v$^1$/+ or Y; R71G10-QF2, QUAS-mtdT:3xHA/P{y[+t7.7] v [+t1.8]=TRiP.HMC04017}attP40; Repo-Gal4 UAS-CD8::GFP/+ | UAS-ball-RNAi (BL#55330) |
| Fig 3F | y$^1$ sc* v$^1$ sev$^{21}$/+ or Y; R71G10-QF2, QUAS-mtdT:3xHA/P{y [+t7.7] v[+t1.8]=TRiP.HMS01723}attP40; Repo-Gal4 UAS-CD8:: GFP/+ | UAS-Syx13-RNAi (BL#38525) |
| Fig 3G | y$^1$ sc* v$^1$ sev$^{21}$/+ or Y; R71G10-QF2, QUAS-mtdT:3xHA/P{y [+t7.7] v[+t1.8]=TRiP.HMS01756}attP40; Repo-Gal4 UAS-CD8:: GFP/+ | UAS-Lac-RNAi (BL#38536) |
| Fig 3H | y$^1$ v$^1$/+ or Y; R71G10-QF2, QUAS-mtdT:3xHA/+; P{y[+t7.7] v [+t1.8]=TRiP.JF01306}attP2/Repo-Gal4 UAS-CD8::GFP | UAS-Art1-RNAi (BL#31348) |
| Fig 3I | y$^1$ v$^1$/+ or Y; R71G10-QF2, QUAS-mtdT:3xHA/+; P{y[+t7.7] v [+t1.8]=TRiP.JF01160}attP2/Repo-Gal4 UAS-CD8::GFP | UAS-Swip-1-RNAi (BL#31585) |
| Fig 3J | y$^1$ sc* v$^1$ sev$^{21}$/+ or Y; R71G10-QF2, QUAS-mtdT:3xHA/P{y [+t7.7] v[+t1.8]=TRiP.HMS02429}attP40; Repo-Gal4 UAS-CD8:: GFP/+ | UAS-Klc-RNAi (BL#42597) |
| Fig 4B | y$^1$ sc* v$^1$ sev$^{21}$/+ or Y; R71G10-QF2, QUAS-mtdT:3xHA/P{y [+t7.7] v[+t1.8]=TRiP.HMC04724}attP40; Repo-Gal4 UAS-CD8:: GFP/+ | UAS-CG2712-RNAi (BL#57418) |

| Figure panel | Full genotype | Comment |
|---|---|---|
| Fig 4C | $y^1$ $v^1$/+ or Y; R71G10-QF2, QUAS-mtdT:3xHA/+; P{y[+t7.7] v[+t1.8]=TRiP.JF01336}attP2/Repo-Gal4, UAS-CD8:GFP | UAS-Mmp1-RNAi (BL#31489) |
| Fig 4D | $y^1$ $v^1$/+ or Y; R71G10-QF2, QUAS-mtdT:3xHA/P{y[+t7.7] v[+t1.8]=TRiP.HMJ23143}attP40; Repo-Gal4, UAS-CD8:GFP/+ | UAS-Mmp2-RNAi (BL#61309) |
| Fig 4F | $y^1$ $v^1$/+ or Y; R71G10-QF2, QUAS-mtdT:3xHA/+; P{y[+t7.7] v[+t1.8]=TRiP.JF01355}attP2/+; OK107-Gal4/+ | UAS-luciferase RNAi (BL#31603) |
| Fig 4G | $y^1$ $v^1$/+ or Y; R71G10-QF2, QUAS-mtdT:3xHA/+; P{y[+t7.7] v[+t1.8]=TRiP.JF01336}attP2/+; OK107-Gal4/+ | UAS-Mmp1-RNAi (BL#31489) |
| Fig 4H | $y^1$ $v^1$/+ or Y; R71G10-QF2, QUAS-mtdT:3xHA/P{y[+t7.7] v[+t1.8]=TRiP.HMJ23143}attP40; OK107-Gal4/+ | UAS-Mmp2-RNAi (BL#61309) |
| Fig 4J | $y^1$ sc* $v^1$ $sev^{21}$/+ or Y; R71G10-QF2, QUAS-mtdT:3xHA/P{y[+t7.7] v[+t1.8]=TRiP.HMC04724}attP40; Repo-Gal4 UAS-CD8::GFP/+ | UAS-CG2712-RNAi (BL#57418) |
| Fig 4K | w/+ or Y; R71G10-QF2, QUAS-mtdT:3xHA/+; {UAS-Mmp1.f1$^{\Delta cat}$}1/Repo-Gal4, UAS-CD8:GFP | APM 1044 (Andrea Page-McCaw) |
| Fig 4L | +/+ or Y; R71G10-QF2, QUAS-mtdT:3xHA/UAS-Mmp2-RNAi; Repo-Gal4, UAS-CD8:GFP/+ | UAS-Mmp2-RNAi (VDRC#107888) |
| Fig S3A | R71G10-QF2, QUAS-mtdT:3xHA/+; Repo-Gal4, UAS-CD8:GFP/+ | Canton S (BL#64349) |
| Fig S3B | $y^1$ $v^1$/+ or Y; R71G10-QF2, QUAS-mtdT:3xHA/P{y[+t7.7] v[+t1.8]=TRiP.HMJ23143}attP40; OK107-Gal4/+ | UAS-Mmp2-RNAi (BL#61309) |
| Fig S3C | $y^1$ $v^1$/+ or Y; R71G10-QF2, QUAS-mtdT:3xHA/P{y[+t7.7] v[+t1.8]=TRiP.HMJ23143}attP40; OK107-Gal4/+ | UAS-Mmp2-RNAi (BL#61309) |
| Fig S3D | $y^1$ $v^1$/+ or Y; R71G10-QF2, QUAS-mtdT:3xHA/P{y[+t7.7] v[+t1.8]=TRiP.HMJ23143}attP40; OK107-Gal4/+ | UAS-Mmp2-RNAi (BL#61309) |
| Fig S3E | $y^1$ $v^1$/+ or Y; R71G10-QF2, QUAS-mtdT:3xHA/P{y[+t7.7] v[+t1.8]=TRiP.HMJ23143}attP40; Repo-Gal4, UAS-CD8:GFP/+ | UAS-Mmp2-RNAi (BL#61309) |
| Fig S4A and B | w/$y^1$ $w^{118}$ or Y; R71G10-QF2, QUAS-mtdT:3xHA/+; 57C10-Gal4/+ | $y^1$,$w^{118}$ (BL#6598) |
| Fig S4C | w/$y^1$ $w^{118}$ or Y; UAS-mCD4:GFP/+; TIFR-Gal4/+ | TIFR-Gal4 (BL#90392) |
| Fig S4D | w/$y^1$ $w^{118}$ or Y; UAS-mCD4:GFP/+; TIFR-Gal4/P{y[+t7.7] v[+t1.8]=TRiP.JF01336}attP2 | UAS-Mmp1-RNAi (BL#31489) |
| Fig S5B | y,w,hsFlp122, UAS-CD8::GFP/y,w or Y; R71G10-Gal4,G13,Gal80/40A,G13,cn,bw | G13 and 40A are FRTs on 2R and 3R, respectively |
| Fig S5C | y,w,hsFlp122, UAS-CD8::GFP/y,w or Y; R71G10-Gal4,G13,Gal80/40A,G13,$mmp2^{\Delta54-120.C}$,cn,bw | G13 and 40A are FRTs on 2R and 3R, respectively |
| Fig S6A | $y^1$ $v^1$/+ or Y; R71G10-QF2, QUAS-mtdT:3xHA/+; P{y[+t7.7] v[+t1.8]=TRiP.JF01355}attP2/+; OK107-Gal4/+ | UAS-luciferase RNAi (BL#31603) |
| Fig S6B | $y^1$ $v^1$/+ or Y; R71G10-QF2, QUAS-mtdT:3xHA/+; {UAS-Mmp1.f1$^{\Delta cat}$}1/+; OK107-Gal4/+ | APM 1044 (Andrea Page-McCaw) |
| Fig S6C | $y^1$ $v^1$/+ or Y; R71G10-QF2, QUAS-mtdT:3xHA/UAS-Mmp2-RNAi; OK107-Gal4/+ | UAS-Mmp2-RNAi (VDRC#107888) |
| Fig S7A | $y^1$ sc* $v^1$ $sev^{21}$/+ or Y; R71G10-QF2, QUAS-mtdT:3xHA/P{y[+t7.7] v[+t1.8]=TRiP.HMC04724}attP40; Repo-Gal4 UAS-CD8::GFP/+ | UAS-CG2712-RNAi (BL#57418) |
| Fig S7B | w[*]/+ or Y; R71G10-QF2, QUAS-mtdT:3xHA/+; P{w[+mC]=UAS-Timp.P}3/Repo-Gal4, UAS-CD8:GFP | UAS-TIMP (BL#58708) |
| Fig S7C | $y^1$ $v^1$/+ or Y; R71G10-QF2, QUAS-mtdT:3xHA/+; P{y[+t7.7] v[+t1.8]=TRiP.JF01355}attP2/+; OK107-Gal4/+ | UAS-luciferase RNAi (BL#31603) |
| Fig S7D | w[*]/+ or Y; R71G10-QF2, QUAS-mtdT:3xHA/+; +/P{w[+mC]=UAS-Timp.P}3; OK107-Gal4/+ | UAS-TIMP (BL#58708) |

**Table 2.  Continued**

| Figure panel | Full genotype | Comment |
|---|---|---|
| Figs S8A and S10A | y[1] v[1]/+ or Y; R71G10-QF2, QUAS-mtdT:3xHA/+; P{y[+t7.7] v[+t1.8]=TRiP.JF01355}attP2/+; OK107-Gal4/+ | UAS-luciferase RNAi (BL#31603) |
| Figs S8B and S10B | y[1] v[1]/+ or Y; R71G10-QF2, QUAS-mtdT:3xHA/+; P{y[+t7.7] v[+t1.8]=TRiP.JF01336}attP2/+; OK107-Gal4/+ | UAS-Mmp1-RNAi (BL#31489) |
| Figs S8C and S10C | y[1] v[1]/+ or Y; R71G10-QF2, QUAS-mtdT:3xHA/P{y[+t7.7] v[+t1.8]=TRiP.HMJ23143}attP40; OK107-Gal4/+ | UAS-Mmp2-RNAi (BL#61309) |
| Figs S8D and S10D | y[1] v[1]/+ or Y; R71G10-QF2, QUAS-mtdT:3xHA/+; P{y[+t7.7] v[+t1.8]=TRiP.JF01355}attP2/Repo-Gal4, UAS-CD8:GFP | UAS-luciferase RNAi (BL#31603) |
| Figs S8E and S10E | y[1] v[1]/+ or Y; R71G10-QF2, QUAS-mtdT:3xHA/+; P{y[+t7.7] v[+t1.8]=TRiP.JF01336}attP2/Repo-Gal4, UAS-CD8:GFP | UAS-Mmp1-RNAi (BL#31489) |
| Figs S8F and S10F | y[1] v[1]/+ or Y; R71G10-QF2, QUAS-mtdT:3xHA/P{y[+t7.7] v[+t1.8]=TRiP.HMJ23143}attP40; Repo-Gal4, UAS-CD8:GFP/+ | UAS-Mmp2-RNAi (BL#61309) |
| Fig 5A | y[1] v[1]/+ or Y; R71G10-QF2, QUAS-mtdT:3xHA/+; P{y[+t7.7] v[+t1.8]=TRiP.JF01336}attP2/Repo-Gal4, UAS-CD8:GFP | UAS-Mmp1-RNAi (BL#31489) |
| Fig 5B | y[1] v[1]/+ or Y; R71G10-QF2, QUAS-mtdT:3xHA/P{y[+t7.7] v[+t1.8]=TRiP.HMJ23143}attP40; Repo-Gal4, UAS-CD8:GFP/+ | UAS-Mmp2-RNAi (BL#61309) |
| Fig 5C | y[1] v[1]/+ or Y; R71G10-QF2, QUAS-mtdT:3xHA/P{y[+t7.7] v[+t1.8]=TRiP.HMJ23143}attP40; P{y[+t7.7] v[+t1.8]=TRiP.JF01336}attP2/Repo-Gal4, UAS-CD8:GFP | UAS-Mmp1-RNAi (BL#31489) + UAS-Mmp2-RNAi (BL#61309) |
| Fig 5E | y[1] v[1]/+ or Y; R71G10-QF2, QUAS-mtdT:3xHA/+; P{y[+t7.7] v[+t1.8]=TRiP.JF01336}attP2/+; OK107-Gal4/+ | UAS-Mmp1-RNAi (BL#31489) |
| Fig 5F | y[1] v[1]/+ or Y; R71G10-QF2, QUAS-mtdT:3xHA/P{y[+t7.7] v[+t1.8]=TRiP.HMJ23143}attP40; OK107-Gal4/+ | UAS-Mmp2-RNAi (BL#61309) |
| Fig 5G | y[1] v[1]/+ or Y; R71G10-QF2, QUAS-mtdT:3xHA/P{y[+t7.7] v[+t1.8]=TRiP.HMJ23143}attP40; P{y[+t7.7] v[+t1.8]=TRiP.JF01336}attP2/+; OK107-Gal4/+ | UAS-Mmp1-RNAi (BL#31489) + UAS-Mmp2-RNAi (BL#61309) |
| Fig 5I | y[1] v[1]/+ or Y; R71G10-QF2, QUAS-mtdT:3xHA/+; P{y[+t7.7] v[+t1.8]=TRiP.JF01336}attP2/Repo-Gal4, UAS-CD8:GFP; OK107-Gal4/+ | UAS-Mmp1-RNAi (BL#31489) |
| Fig 5J | y[1] v[1]/+ or Y; R71G10-QF2, QUAS-mtdT:3xHA/P{y[+t7.7] v[+t1.8]=TRiP.HMJ23143}attP40; Repo-Gal4 UAS-CD8:GFP/+; OK107-Gal4/+ | UAS-Mmp2-RNAi (BL#61309) |
| Fig 5K | y[1] v[1]/+ or Y; R71G10-QF2, QUAS-mtdT:3xHA/P{y[+t7.7] v[+t1.8]=TRiP.HMJ23143}attP40; P{y[+t7.7] v[+t1.8]=TRiP.JF01336}attP2/Repo-Gal4, UAS-CD8:GFP; OK107-Gal4/+ | UAS-Mmp1-RNAi (BL#31489) + UAS-Mmp2-RNAi (BL#61309) |
| Fig S9A | y[1] w[118]/+ or Y; R71G10-QF2, QUAS-mtdT:3xHA/+; +/Alrm-Gal4 UAS-CD8::GFP | y[1],w[118] (BL#6598) |
| Fig S9B | y[1] v[1]/+ or Y; R71G10-QF2, QUAS-mtdT:3xHA/+; P{y[+t7.7] v[+t1.8]=TRiP.JF01336}attP2/Alrm-Gal4, UAS-CD8:GFP | UAS-Mmp1-RNAi (BL#31489) |
| Fig S9C | y[1] v[1]/+ or Y; R71G10-QF2, QUAS-mtdT:3xHA/P{y[+t7.7] v[+t1.8]=TRiP.HMJ23143}attP40; Alrm-Gal4, UAS-CD8:GFP/+ | UAS-Mmp2-RNAi (BL#61309) |

(Bornstein et al, 2015). It is thus possible that MMPs cleave FasII, or other adhesion molecules, to promote axon pruning. Because most *Drosophila* MMP isoforms are secreted (LaFever et al, 2017), together with the finding that they are required from both neuronal and glial origin, it is possible that *MMPs* are also secreted from additional cells within or around the MB circuit. All the above-mentioned options are promising directions for future research, in hopes of uncovering the mechanisms by which *MMPs* promote neuronal remodeling.

Notably, the main hypothesis in the literature is that SCZ is associated with increased synapse pruning, whereas in our study, KD of SCZ-associated genes resulted in inhibition of pruning, which may seem counterintuitive at first. However, SNPs can result in not only loss but also gain of function, which could account for this apparent conflict. In fact, it was previously shown that the SCZ-associated SNP in *Mmp9* is in the 3′ UTR, and affects Mmp9 mRNA folding, which eventually results in higher Mmp9 compared with WT (Lepeta et al, 2017).

We recognize that the current study explores pruning of long stretches of axons, whereas SCZ is mainly associated with defective synapse pruning. More generally, flies could never model the complexities associated with human psychiatric conditions, nor

directly offer therapeutic targets. Nonetheless, this simple and genetically accessible model enables us to uncover conserved neurodevelopmental principles and concepts. Understanding the neurodevelopmental role of *MMPs* may provide insights into brain ECM dynamics throughout development, and specifically during neuronal remodeling.

# Materials and Methods

### Gene selection for the screen

Human genes that contain SNPs associated with SCZ were collected from two studies: PGC2 and CLOZUK (Wu et al, 2017; Table S1). These genes were converted to their *D. melanogaster* homologs using HumanMine (Table S1). In cases of multiple fly homologs for a single human gene, we decided to include all of them.

### *D. melanogaster* rearing and strains

All fly strains were reared under standard laboratory conditions at 25°C on molasses-containing food. Males and females were chosen at random. Unless specifically stated otherwise, the relevant developmental stage is adult, which refers to 3–5 d post-eclosion. The RNA lines used for the screen are detailed in Table 1. The following lines were obtained for the Bloomington *Drosophila* Stock Center (BDSC): luciferase RNAi (used as a control for RNAi experiments; #31603), CG2712 RNAi (used as a control for RNAi experiments; #57418), Mmp1 RNAi (#31489), UAS-TIMP (#58708), Alrm-Gal4 (#67032), Repo-Gal4 (#7415), OK107-Gal4 (#854), TIFR-Gal4 (#90392), QUAS-mtdTomato-3xHA (#30004), Mmp2-gRNA (#82521). R71G10-QF2 was previously generated by our laboratory (Bornstein et al, 2021). R71G10-Gal4 on the second chromosome was previously generated by our laboratory (Alyagor et al, 2018). A second Mmp2 RNAi line was obtained from the Vienna *Drosophila* Resource Center (VDRC; #107888). Line UAS-Mmp1.f1$^{\Delta Cat1}$ was a generous gift from Andrea Page-McCaw. See full genotypes (ordered by specific figure panels) in Table 2.

### Immunohistochemistry and imaging

*Drosophila* brains were dissected in cold Ringer's solution and placed on ice, followed by fixation in 4% PFA solution for 20 min at RT. Fixed brains were then washed with PB supplemented with 0.3% Triton X (PBT)—3 times, 20 min each. Next, the brains were blocked for nonspecific staining using 5% heat-inactivated normal goat serum (NGS). Antibody staining was performed as follows: primary antibodies (4°C, overnight), PBT washes (X3 in RT, 20 min each), secondary antibodies (RT, 2 h), PBT washes (X3 in RT, 20 min each), mounting in SlowFade (Invitrogen). Primary antibodies included mouse anti-FasII 1:25 (1D4; Developmental Studies Hybridoma Bank [DSHB]), mouse anti-EcRB1 1:25 (AD4.4; DSHB), rat anti-HA 1:250 (11867423001; Sigma-Aldrich), and mouse anti-Mmp1 1:20 (mix of 3A6B4, 3B8D12, and 5H7B11; DSHB). Secondary antibodies included Alexa Fluor 647 goat anti-mouse 1:300 (A-21236; Invitrogen) and

Alexa Fluor 568 goat anti-rat 1:300 (A-21247; Invitrogen). All brains were imaged on a Zeiss LSM 980 confocal microscope. The images were processed using ImageJ 2.14.0 (NIH).

### Generation of the mmp2 mutant allele

The *mmp2* mutant allele named Δ54–120.C was generated by crossing a gRNA line, which harbors two distinct gRNA sequences both targeting the 5′ CDS region common to all 3 Mmp2 isoforms (cloned into pCFD4; BDSC #82521, WKO.3-C7), to a nanos-Cas9–expressing fly. The recovered indel is a deletion of 67 bp (54–120 in the CDS of isoform C)—predicted to induce a premature stop codon (after 77 out of 606 amino acids in isoform C). Unfortunately, this mutant (alongside many others) was lost on June 15 (see the Acknowledgements section).

The following primers were used to sequence the indel mutation:
F: GCATTCAATGCTGCCACAAA.
R: CATTTCATCATCGACGTCGT.

### Generation of MARCM clones

γ-KC neuroblast clones were generated using the MARCM strategy (Lee et al, 1999). Newly hatched larvae (~24 h after egg laying) were heat-shocked for 1 h at 37°C. Adult brains were dissected for further analysis.

### Quantification and statistical analysis

Blind ranking of the pruning phenotypes for Figs 4 and 5 was performed, for all experiments, by two independent investigators who reviewed the Z-projections: ranking scores: 1 = WT-like phenotype; 2 = very mild pruning defect (PD); 3 = mild PD; 4 = moderate PD; 5 = severe PD (Fig S3A–E). The two sets of ranking were compared using the Wilcoxon signed-rank test and found to be statistically insignificant (Fig S3F and G).

A nonparametric statistical analysis was performed to analyze the ranking (Figs 4 and 5): the Kruskal–Wallis H test, followed by a post hoc Mann–Whitney *U* test with FDR correction.

*** = $P < 0.001$; ** = $P < 0.01$; * = $P < 0.05$; n.s. = not significant. Specific *P*-values are indicated in the figure legends.

# Data Availability

Raw data are available upon request. Unfortunately, new fly lines made in this work have been destroyed (see the Acknowledgements section) and therefore cannot be disseminated.

# Supplementary Information

# Acknowledgements

We thank Andrea Page-McCaw (Vanderbilt University) for kindly sharing the Mmp1 dominant negative strain, and the Vienna *Drosophila* Resource Center and the Bloomington *Drosophila* Stock Center for reagents. Monoclonal antibodies were obtained from the Developmental Studies Hybridoma Bank developed under the auspices of the NICHD and maintained by the University of Iowa. We thank R Rothkopf for assistance with statistics. This work was supported by European Research Council (ERC) advanced grant #101054886 "NeuRemodelBehavior," and by the ISF-NSFC Joint Research Program (grant #2573/18). O Schuldiner is an incumbent of Prof. Erwin Netter Professorial Chair of Cell Biology. On the night of June 15, 2025, our laboratory was directly hit by an Iranian missile and was completely destroyed. Unfortunately, this affects our ability to provide reagents and limited the scope of our revisions.

## Author Contributions

S Keret: conceptualization, data curation, formal analysis, investigation, methodology, and writing—original draft, review, and editing.
H Meltzer: conceptualization, supervision, investigation, methodology, and writing—original draft, review, and editing.
N Marmor-Kollet: conceptualization.
O Schuldiner: conceptualization, supervision, funding acquisition, project administration, and writing—original draft, review, and editing.

## Conflict of Interest Statement

The authors declare that they have no conflict of interest.

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
