## [Reviewer comments · Life Science Alliance]

A Drosophila screen of schizophrenia genes highlights neural and glial MMPs in neuronal remodeling

Shir Keret, Hagar Meltzer, Neta Marmor-Kollet, and Oren Schuldiner

DOI: <https://doi.org/10.26508/lsa.202503504>

Corresponding author(s): Oren Schuldiner, Weizmann Institute of Science and Hagar Meltzer, Weizmann Institute

Review Timeline:

Submission Date:	2025-09-08
Editorial Decision:	2025-09-09
Revision Received:	2025-09-11
Editorial Decision:	2025-09-16
Revision Received:	2025-09-17
Accepted:	2025-09-17

Scientific Editor: Tim Fessenden

Transaction Report:

Please note that the manuscript was reviewed at *Review Commons* and these reports were taken into account in the decision-making process at *Life Science Alliance*.

Reviews

Review #1

1. Evidence, reproducibility and clarity:

****Summary:****

This study reports a role for matrix metalloproteinases (MMPs) in the developmental pruning of gamma Kenyon cells (KCs) in the fruit fly Mushroom Body during larval-pupal metamorphosis. The authors show through gene expression studies that MMP genes are upregulated in late larval stages as part of the early program for this type of neuronal pruning. They show through cell-targeted RNAi studies of both secreted MMP-1 and membrane-anchored MMP-2, that both genes are required in glial cells and to a lesser extent within KCs. The authors show that MMP secreted from glial is required for normal levels of Mushroom Body developmental neuronal pruning. They mention that MMP genes have been identified in schizophrenic patient screens in patients, and that perhaps a comparable pruning mechanism could be involved in the loss of grey matter (loss of synapses) in patients. The authors propose that MMP levels may be a potential therapeutic marker in the future.

****Major Comments:****

Overall, the work is of a reasonable standard, but very preliminary. The study lacks the substance to completely convince me of any of the results. There is SUBSTANTIAL work that needs to be done to make this publishable. There are a lot of writing mistakes; so many that I do not list them in detail here. The references citations are fairly old, but I do not list update replacements here. The text is very brief, and the overall writing needs to include significantly more description and detail. This is evident in all aspects of the manuscript, but especially notable in the Methods and Figure Legends.

None of the Figure Legends include full genotypes of any of the fly lines, and these full fly lines are also not included in the Methods. This is vital to compare the experimental lines to the controls. Major points are listed below:

1. Figure 2: It is important to note of the specific age of animals in these images when talking about the loss of genes in development. Are all the animals age-matched? High levels of synaptic pruning occur post-eclosion), and it is important to understand when these pruning defects occur. It is mentioned that that overlap for the gene expression data is upregulated during 6-18h APF is this when these images are taken? This is very important in the context of pruning as SCZ symptom presentation is very late relative to these early events.
2. Figure 2: In the figure legend, it is indicated that the arrows are unpruned axons, however in the controls these areas appear to be highly innervated. Further explanation is needed about the context of the arrows, as there are clear visual differences between these images and the controls, but they appear to have a more expansive phenotype than "unpruned axons". The data does not match the visual representation in comparison to the control.
3. Figure 2: There needs to be more descriptive definitions and clarifications to the defects labeled in panel K. This could be done in the figure legend, but it would be more useful to label the images provided. For example, if *Mmp2* is a "mild pruning affect, put that in the pie chart somewhere, to help guide the description of the phenotype to what those confocal images look like.
4. Figure 3: The time points of the images of the Mushroom Body (MB) are vital to understanding the process and regulation of these genes.
5. Figure 3D: Significant description of this graph needs to be added for clarity. What parameters separate each phenotypic defect? Labeling the images and showing images that belong in different groups would be very helpful and improve the paper significantly.
6. Figure S1: Additional experiments would help answer the strength of the phenotype for the ALG-Gal 4 driver. The authors need to perform the rescue experiment. Use a MMP-2 null and then drive it back in the ALG-GAL4 to see if this is sufficient to rescue the neuron pruning. This also isolates the mechanisms to one subtype of glia.
7. Figure 3 and 4: The other glial subtypes need to be analyze to make any conclusion about their involvement, as well as the involvement of the astrocytes. Running these exact same experiments on the cortex glial and ensheathing glia will provide essential insight into what glial subtype is involved. The presumed lack of phenotypes in these other

glial subtypes will also strengthen the argument that the astrocytes are specifically involved in this process. These are vital experiments.

8. Figure 4: Again, description of the phenotypes and examples of these would improve the quality of this figure substantially.

9. Figure 5: An improvement on the quantifications of these phenotypes would strengthen the paper substantially. More detailed description of the phenotypes and how they related to the control would significantly improve the overall quality of the work.

****Minor Comments:****

1. Figure 1A: I would suggest labeling the KC (γ) and potentially one of the others (a/B, a'/B') to orient the reader to the differences between these two subsets of the KCs, and to emphasize which neurons are undergoing pruning and where the cell bodies are and where the axons project.

2. Figure 1C: This panel needs further labeling to explain the findings in the heat map. Labeling some of the genes that were found and where they were would be helpful. This could also be done in the figure legend, however without any further labeling or context the heatmap is confusing.

3. Figure 3B,C: The full genotypes need to be labeled. What is the exact genotype used for the control?

4. Figure S1: The stock number for the ALG-GAL4 is missing, there are multiple different drivers, therefore this could be helpful in understanding this phenotype, as some are better than others.

5. Figures 3 and 4: Labeling needs to remain consistent; Figure 3 "Glia-Gal4", Figure 4 "glia-gal4".

2. Significance:

Significance (Required)

***General Assessment:** An interesting study on MMP function during an unusual type of neural development (axon pruning). Most of the MMP function appears to be in glia, although the MMP role in this context is unclear. The MMP function in the neurons being pruned is unexpected and even less clear. The study is somewhat poorly described in terse language lacking essential information, which gives the overall impression of a preliminary report.

***Advance:** Glial MMP function has been described for neuronal clearance mechanisms following injury. The main advance here is to describe a similar function during normal development.

***Audience:** Developmental neuroscientists, MMP biologists, possibly schizophrenia clinician researchers

3. How much time do you estimate the authors will need to complete the suggested revisions:

Estimated time to Complete Revisions (Required)

(Decision Recommendation)

Between 3 and 6 months

4. Review Commons values the work of reviewers and encourages them to get credit for their work. Select 'Yes' below to register your reviewing activity at Web of Science Reviewer Recognition Service (formerly Publons); note that the content of your review will not be visible on Web of Science.

No

Review #2

1. Evidence, reproducibility and clarity:

Neuropsychiatric conditions are often influenced by genetic factors. Schizophrenia is a complex mental disorder characterised by a mixture of hallucinations, delusions and disorganised thinking that causes lifelong problems in daily life. GWAS have identified a number of genes associated with the risk of developing schizophrenia, although genetic predisposition alone is not sufficient and additional environmental factors are required.

In the current manuscript, the authors aim to exploit the strength of the *Drosophila* system to explore a link between schizophrenia-associated genes and neuronal remodelling during development. They focus on the mushroom body in the adult brain, where pronounced neuronal remodelling occurs during metamorphosis. To assess the potential role of the genes identified by the GWAS, they performed a targeted RNAi-based screen. They focus on the role of metalloproteases and find that they are required in neurons and in glia for the pruning of mushroom body axons.

The study starts with a selection of 32 genes, 29 of which are listed (a bit hidden) in materials and methods and the identification of the *Drosophila* orthologs. The expression patterns of these genes in Kenyon cells are presented in Figure 1 - but unfortunately no information is given on who is expressed when. In a next step, Kenyon cell specific RNAi knockdown experiments are shown that identify a pruning phenotype for several genes. They demonstrate that Mmp2 (and similarly Mmp1) is also required in glia. Although Mmp2 was identified by neuronal RNAi-based knockdown, double knockdown experiments led the authors conclude that its primary function is in glia.

The study emphasises the use of the advanced genetic model to understand complex human diseases. However, the paper does not go far enough in making use of the excellent genetics available. Basically, the report is about the identification of a few hits in a small RNAi screen, which is fine in itself, but leaves many questions unanswered. Do mmp1/2 mutants have a phenotype? Can the phenotype be rescued? Does TIMP expression lead to similar phenotypes? What is the temporal requirement for Mmp1/2? What are the target proteins of Mmp2? Is Mmp2 still required when astrocyte motility is blocked? What is the morphology of glia after Mmp1/2 knockdown?

2. Significance:

Significance (Required)

The strength of the study is to identify a pruning phenotype after RNAi-based knockdown. The limitations is that this study is very superficial, it is the beginning of a paper. The initial claim to use *Drosophila* because to its advanced genetics is not met. The results section is shorter than the discussion.

3. How much time do you estimate the authors will need to complete the suggested revisions:

Estimated time to Complete Revisions (Required)

(Decision Recommendation)

Cannot tell / Not applicable

4. Review Commons values the work of reviewers and encourages them to get credit for their work. Select 'Yes' below to register your reviewing activity at Web of Science Reviewer Recognition Service (formerly Publons); note that the content of your review will not be visible on Web of Science.

Yes

Review #3

1. Evidence, reproducibility and clarity:

In this work, Schuldiner and colleagues explore the role of Mmp1 and Mmp2 in neuronal remodeling in the mushroom body of *Drosophila*. Overall, this work is very interesting, but in its current form seems quite preliminary. The biggest limitation of the study is that single RNAi lines are used with no validation that the lines are working, despite the fact that Mmp antibodies are available as are endogenously tagged Mmp lines that could have been used to validate the genetic manipulations. Specific concerns are listed below.

****Major concerns****

1. The scoring system for pruning of mushroom body neurons seems very variable, even in controls (where scoring can range from very mild to moderate), and it is very hard to assess from the images what one is looking at (rather than using our own judgment, we rely on the authors' words). It would be necessary to have better labeling and examples of what phenotypes are considered "mild", "severe", "wild type-like". It would also help to understand how phenotype assessment is guided by the overlap between the signals from TdTomato fluorescence and FasII stain.

2. The biggest limitations of the approach are that single RNAi lines are used to screen, with no accompanying validation of the tool (see above)

3. RNAi-based knockdown is used to infer epistatic information-this is not appropriate as epistasis experiments need to be done with null alleles to make firm conclusions. Additional concerns:

- Even with the same driver, knockdown efficiency for 2 different genes could be variable and dependent of the specific RNAi used.

- The comparison between drivers is even harder, as driver strength varies greatly.

- The knockdown efficiency drops with increasing numbers of RNAi used.

- The specific genotypes used for this experiment should be clarified, as it would be very important to ensure that the UAS dosage is equal across conditions.

4. To further deepen the rigor of this work, a few simple yet important things could have been done. First, it would be important to rule out that knocking down Mmps does not affect astrocyte numbers and health (could be assessed by counting numbers and observing their morphology). Also, the authors previously showed that astrocytes actively infiltrate the axon bundle prior to pruning to facilitate axon defasciculation and pruning (Marmor-Kollet et al., 2023). It would have provided an important insight to examine if astrocytes can infiltrate the axon bundle if Mmp2 and/or Mmp1 are knocked down.

****Minor****

The introduction puts big emphasis on the role of glia, but then to narrows down candidate genes for the screen a γ -KCs transcriptional data set is used, and the initial screen is done via knockdown of those candidates in neurons (there is a disconnect between rationale and approach).

Rationale for looking into axon pruning and how that translates into insights about synaptic pruning defects in schizophrenia should be more clearly stated.

Figure 1C: data visualization for this heat map should be improved. Parts of the data are faded, and the differences between gene clusters are unclear.

2. Significance:

Significance (Required)

In this work, Schuldiner and colleagues explore the role of Mmp1 and Mmp2 in neuronal remodeling in the mushroom body of *Drosophila*. Overall, this work is very interesting, but in its current form seems quite preliminary. The biggest limitation of the study is that single RNAi lines are used with no validation that the lines are working, despite the fact that Mmp antibodies are available as are endogenously tagged Mmp lines that could have been used to validate the genetic manipulations.

3. How much time do you estimate the authors will need to complete the suggested revisions:

Estimated time to Complete Revisions (Required)

(Decision Recommendation)

Between 3 and 6 months

4. Review Commons values the work of reviewers and encourages them to get credit for their work. Select 'Yes' below to register your reviewing activity at Web of Science Reviewer Recognition Service (formerly Publons); note that the content of your review will not be visible on Web of Science.

No

September 9, 2025

Re: Life Science Alliance manuscript #LSA-2025-03504-T

Prof. Oren Schuldiner
Weizmann institute
Molecular Cell Biology
Herzl 234
Rehovot, NA 7610001
Israel

Dear Dr. Schuldiner,

Thank you for transferring your manuscript entitled "A *Drosophila* screen of schizophrenia-related genes highlights MMP requirement in neuronal remodeling" to Life Science Alliance.

In accordance with our offer to publish this work, please include the new data on TIMP in the manuscript as a supplemental figure, with the appropriate caveats about its interpretation. Please note that once we receive the revised manuscript we will check it for any final formatting changes needed to meet LSA's requirements.

Thank you for this interesting contribution to Life Science Alliance. We are looking forward to receiving your revised manuscript.

Sincerely,

B. MANUSCRIPT ORGANIZATION AND FORMATTING:

spreadsheets for the main figures of the manuscript. If you would like to add source data, we would welcome one PDF/Excel-file per figure for this information. These files will be linked as supplementary "Source Data" files.

Manuscript number: RC-2024-02721

Corresponding authors: Hagar Meltzer, Oren Schuldiner

[Please use this template only if the submitted manuscript should be considered by the affiliate journal as a full revision in response to the points raised by the reviewers.]

*If you wish to submit a preliminary revision with a revision plan, please use our "Revision Plan" template. **It is important to use the appropriate template to clearly inform the editors of your intentions.**]*

1. General Statements [optional]

Our lab was totally destroyed on June 15th by an Iranian missile. All stocks, equipment and reagents were lost. While we performed many of the experiments requested by the reviewers, unfortunately some were never completed. We thank you for your understanding.

We thank the three reviewers for their thoughtful comments and useful suggestions on how to improve our paper. Some of the reviewers claimed that the paper is "preliminary". We would like to highlight that in our opinion "preliminary" has two possible meanings in this context: 1) the data does not yet support the claims that the authors wrote; 2) the story is short and should be extended. While we totally agree that type 1 "preliminary" should be addressed (and we have addressed that to the best of our abilities), type 2 "preliminary" is a matter of scope, the length of the paper/project and the publication home. We believe that this story, which has been led by an outstanding master's student (and as such has had a limited timespan) is worthwhile of publication in its current scope.

*Reviewers' comments are in **BLUE** while our responses are in **BLACK**.*

Reviewer 1 Summary:

This study reports a role for matrix metalloproteinases (MMPs) in the developmental pruning of gamma Kenyon cells (KCs) in the fruit fly Mushroom Body during larval-pupal metamorphosis. The authors show through gene expression studies that MMP genes are upregulated in late larval stages as part of the early program for this type of neuronal pruning. They show through cell-targeted RNAi studies of both secreted MMP-1 and membrane-anchored MMP-2, that both genes are required in glial cells and to a lesser extent within KCs [comment – both MMPs have secreted and membrane-anchored isoforms and we did not assess whether the secreted/anchored isoforms are involved; e.g. see LaFever et al. 2017]. The authors show that MMP secreted from glial is required for normal levels of Mushroom Body developmental neuronal pruning. They mention that MMP genes have been identified in schizophrenic patient

screens in patients, and that perhaps a comparable pruning mechanism could be involved in the loss of grey matter (loss of synapses) in patients. The authors propose that MMP levels may be a potential therapeutic marker in the future.

We thank the reviewer for his comments. We find it important to clarify that we do not think our work suggests that the MMPs levels may be a potential therapeutic marker without much additional work in the future. In the original text we added a claim from another paper suggesting MMPs as therapeutic target. However, due to the arising confusion, we decided to delete this statement from the text (original line 198). We also added a general disclaimer towards the end of the discussion regarding the genetic power of *Drosophila* but its limited implication into human health (new lines 276-278).

Major Comments:

Overall, the work is of a reasonable standard, but very preliminary [please see general note on two types of “preliminary” – we thank the reviewer for helping us substantiate our claims and strengthen our paper but we do not plan to significantly increase its scope]. The study lacks the substance to completely convince me of any of the results. There is SUBSTANTIAL work that needs to be done to make this publishable. There are a lot of writing mistakes; so many that I do not list them in detail here [we are not absolutely sure that we understand to which mistakes this reviewer is eluding. However, we carefully rewrote the manuscript, streamlined many of our claims and added many new and more recent references]. The references citations are fairly old, but I do not list update replacements here [Thanks – we added many newer and relevant citations]. The text is very brief, and the overall writing needs to include significantly more description and detail [we have included more descriptions and details, as will be elaborated later on, but – again - this is a short report and will remain as such]. This is evident in all aspects of the manuscript, but especially notable in the Methods and Figure Legends [Thanks for raising this comment, which was reverberated also by other reviewers – we have now included more details, with a particular focus on the genotypes (Table 2), that somehow were erroneously not included in the original submission, as well as more detailed figure legends].

None of the Figure Legends include full genotypes of any of the fly lines, and these full fly lines are also not included in the Methods. This is vital to compare the experimental lines to the controls [true – our apologies for this mistake, we now added the full genotypes in Table 2].

Major points are listed below:

1. Figure 2: It is important to note of the specific age of animals in these images when talking about the loss of genes in development. Are all the animals age-matched? High levels of synaptic pruning occur post-eclosion), and it is important to understand when these pruning defects occur. It is mentioned that that overlap for the gene expression data is upregulated during 6-18h APF is this when these images are taken? This is very important in the context of pruning as SCZ symptom presentation is very late relative to these early events.

We thank the reviewer for this comment which suggests we were not clear enough in our description. We do not claim to have generated an SCZ model and have clarified this better in the text (lines 275-278). Furthermore, axon pruning happens during pupal development, but in all the main figures in this manuscript we dissected young adult flies (3-5 days post eclosion)

and show the remnants of unpruned axons (as we have done in numerous studies). To make sure that initial development occurred normally, we also include larval brains in the Figure S8. We now clarified the fact that we are imaging adult brains as a readout to investigate whether pruning occurred during metamorphosis or not (line 124-126).

2. Figure 2: In the figure legend, it is indicated that the arrows are unpruned axons, however in the controls these areas appear to be highly innervated. Further explanation is needed about the context of the arrows, as there are clear visual differences between these images and the controls, but they appear to have a more expansive phenotype than "unpruned axons". The data does not match the visual representation in comparison to the control.

We apologize for this confusion. Unfortunately, the driver which we use to label the γ -axons, R71G10-QF2, is not absolutely specific to the γ type KCs but also expressed (sometimes) in the α/β KCs. As the α/β axons are very stereotypic in shape and also express high levels of FasII (which we stain for), we can easily distinguish between the α lobe and unpruned γ axons. To clarify this point, we now clearly demarcate all lobes in the control images and specifically the α lobe in all panels. Additionally, we added new schemes in Figure 2A and 2O to better clarify the anatomy and experimental design.

3. Figure 2: There needs to be more descriptive definitions and clarifications to the defects labeled in panel K. This could be done in the figure legend, but it would be more useful to label the images provided. For example, if Mmp2 is a "mild pruning affect, put that in the pie chart somewhere, to help guide the description of the phenotype to what those confocal images look like.

We understand that the pie chart in Figure 2 was confusing and therefore simplified it in the current version (Fig. 2B and 2P). Also, thanks to this great point, we now include a new Figure S3 that includes examples for the ranking categories, which were now performed by two independent investigators in a blind manner.

4. Figure 3: The time points of the images of the Mushroom Body (MB) are vital to understanding the process and regulation of these genes.

Please see our comment to point #1 – unless specifically stated otherwise, all images are MBs of adult flies, as now clearly mentioned in the figure legends, in the text and in the Material and Methods section.

5. Figure 3D: Significant description of this graph needs to be added for clarity. What parameters separate each phenotypic defect? Labeling the images and showing images that belong in different groups would be very helpful and improve the paper significantly.

We now included a new Figure S3 (also see our response to comment #3).

6. Figure S1: Additional experiments would help answer the strength of the phenotype for the ALG-Gal 4 driver. The authors need to perform the rescue experiment. Use a MMP-2 null and then drive it back in the ALG-GAL4 to see if this is sufficient to rescue the neuron pruning. This also isolates the mechanisms to one subtype of glia.

These are excellent suggestions that are, unfortunately, not doable. To perform a rescue experiment, one would need a viable loss-of-function phenotype of an Mmp2 mutant. There is one published Mmp2 loss-of-function null allele which is lethal during pupal development (Page-McCaw et al, 2003). Our previous data, using tissue specific (ts)CRISPR, suggested the involvement of Mmp2 in neurons for their remodeling (Meltzer et al, 2019). We therefore

independently generated an Mmp2 germline mutant using CRISPR (harboring an indel resulting in a premature stop codon and predicted to encode a truncated, 77 amino-acid long protein), now described in Fig. S5A (and in the Materials and Methods). This allele is, as expected, unfortunately also lethal. We attempted to overcome lethality by generating MARCM (mosaic) clones in neurons, but as expected, because Mmp2 is largely secreted, there was no pruning defect phenotype (Fig. S5B-C). A similar problem would also be relevant to glial MARCM clones (as the non-clonal glial population would still secrete the MMPs).

7. Figure 3 and 4: The other glial subtypes need to be analyzed to make any conclusion about their involvement, as well as the involvement of the astrocytes. Running these exact same experiments on the cortex glial and ensheathing glia will provide essential insight into what glial subtype is involved. The presumed lack of phenotypes in these other glial subtypes will also strengthen the argument that the astrocytes are specifically involved in this process. These are vital experiments.

We currently limited our analysis (and conclusions) to astrocytes. Despite the fact that this experiment is beyond our initial scope, we obtained reagents and performed preliminary experiments (using the R77A03-Gal4 driver for cortex glia, and the R83E12-Gal4 for ensheathing glia). In both cases, we observed extremely mild pruning defects, not comparable to those with Repo- or Alrm-Gal4. In these preliminary experiments we lacked a proper control, and now, unfortunately, due to the loss of our lab, we are unable to complete these experiments in a reasonable amount of time.

8. Figure 4: Again, description of the phenotypes and examples of these would improve the quality of this figure substantially.

Absolutely agree – see our response to comment #3 (and Fig. S3).

9. Figure 5: An improvement on the quantifications of these phenotypes would strengthen the paper substantially. More detailed description of the phenotypes and how they related to the control would significantly improve the overall quality of the work.

Thanks again for highlighting that we neglected to include the full genotypes that are now added (Table 2). We also thank the reviewer for raising the point regarding quantification. First, we generated a new Fig. S3A-E to show examples of the ranking by two independent rankers. Second, ranking was performed by looking at TdTomato positive vertical axons that are outside of the α lobe (high FasII) – this is now better explained in the materials and methods. Additionally, while we would love to have a better scoring, and automatic, system – and even published a semi-automated scoring algorithm in Alyagor et al. 2018 (Figure 3O in the Alyagor paper), because the driver also labels vertical axons (α/β) and because unpruned γ axons often express FasII, this quantification method does not always work. What we have done in previous cases, as we have also done here, is to provide independent ranking by two investigators and compare their ranking (Fig. S3F-G). Finally, we are working with our AI hub to develop automatic scoring systems that will not require human ranking – however this is beyond the scope for this manuscript.

Minor Comments:

1. Figure 1A: I would suggest labeling the KC (gamma) and potentially one of the others (a/B, a'/B') to orient the reader to the differences between these two subsets of the KCs, and to

emphasize which neurons are undergoing pruning and where the cell bodies are and where the axons project.

Thanks for the suggestions – we now better annotated the scheme in Figure 1A as well as additional schematics in Figure 2 and, finally, better annotations in selected panels. Specifically, the α lobe is outlined in magenta throughout all relevant panels.

2. Figure 1C: This panel needs further labeling to explain the findings in the heat map. Labeling some of the genes that were found and where they were would be helpful. This could also be done in the figure legend, however without any further labeling or context the heatmap is confusing.

We apologize for the incomplete figure. We did not want to overload the figure with data, which is why we are showing only the important clusters and did not include gene names. To keep the figure simple, but at the same time provide the complete information, we now include the full data in Fig. S1 (that includes the original heatmap with all the dynamic clusters I-IX, and including all the gene names). For the full raw data, including non-dynamic clusters, the reader is referred to look in Supplemental excel file 1. We hope this provides the clarity that this reviewer rightfully asks for.

3. Figure 3B,C: The full genotypes need to be labeled. What is the exact genotype used for the control?

The full genotypes of all figure panels are now included in Table 2 in the Materials and Methods.

4. Figure S1: The stock number for the ALG-GAL4 is missing, there are multiple different drivers, therefore this could be helpful in understanding this phenotype, as some are better than others.

Indeed, *Alm-Gal4* comes on two chromosomes – we used BDSC #67032, which is on chromosome III and this is now clearly mentioned in the Materials and Methods section.

5. Figures 3 and 4: Labeling needs to remain consistent; Figure 3 "*Glia-Gal4*", Figure 4 "*glia-gal4*".

Thanks, done.

Reviewer #1 (Significance (Required)):

General Assessment: An interesting study on MMP function during an unusual type of neural development (axon pruning). Most of the MMP function appears to be in glia, although the MMP role in this context is unclear. The MMP function in the neurons being pruned is unexpected and even less clear. The study is somewhat poorly described in terse language lacking essential information, which gives the overall impression of a preliminary report.

Advance: Glial MMP function has been described for neuronal clearance mechanisms following injury. The main advance here is to describe a similar function during normal development.

Audience: Developmental neuroscientists, MMP biologists, possibly schizophrenia clinician researchers

Reviewer #2 (Evidence, reproducibility and clarity (Required)):

Neuropsychiatric conditions are often influenced by genetic factors. Schizophrenia is a complex mental disorder characterised by a mixture of hallucinations, delusions and disorganised thinking that causes lifelong problems in daily life. GWAS have identified a number of genes associated with the risk of developing schizophrenia, although genetic predisposition alone is not sufficient and additional environmental factors are required.

In the current manuscript, the authors aim to exploit the strength of the *Drosophila* system to explore a link between schizophrenia-associated genes and neuronal remodelling during development. They focus on the mushroom body in the adult brain, where pronounced neuronal remodelling occurs during metamorphosis. To assess the potential role of the genes identified by the GWAS, they performed a targeted RNAi-based screen. They focus on the role of metalloproteases and find that they are required in neurons and in glia for the pruning of mushroom body axons.

The study starts with a selection of 32 genes, 29 of which are listed (a bit hidden) in materials and methods and the identification of the *Drosophila* orthologs. The expression patterns of these genes in Kenyon cells are presented in Figure 1 - but unfortunately no information is given on who is expressed when [we apologize for the confusion. We attempted to keep Figure 1 simple but this resulted in the absence of critical information, as the reviewer suggests. We now include a Figure S1 that includes the entire heatmap of the dynamically expressed clusters I-IX with all the gene names. Additionally, we now augmented the information in Table 1 to include the screen phenotypes. Finally, Supplemental excel file 1, also included in our original submission, includes all the data, and is now better referred to throughout the text]. In a next step, Kenyon cell specific RNAi knockdown experiments are shown that identify a pruning phenotype for several genes. They demonstrate that *Mmp2* (and similarly *Mmp1*) is also required in glia. Although *Mmp2* was identified by neuronal RNAi-based knockdown, double knockdown experiments led the authors conclude that its primary function is in glia.

The study emphasises the use of the advanced genetic model to understand complex human diseases. However, the paper does not go far enough in making use of the excellent genetics available. Basically, the report is about the identification of a few hits in a small RNAi screen, which is fine in itself, but leaves many questions unanswered. Do *mmp1/2* mutants have a phenotype?

This is a very important question that cannot be answered, unfortunately. There is one published *Mmp2* loss of function null allele which is lethal during pupal development (Page-MaCaw et al, 2003). Our previous data, using tissue specific (ts)CRISPR, suggested the involvement of *Mmp2* in neurons for their remodeling (Meltzer et al, 2019). We therefore independently generated an *Mmp2* germline mutant using CRISPR (harboring an indel resulting in a premature stop codon and predicted to encode a truncated, 77 amino-acid long protein), now described in Fig. S5A (and in the Materials and Methods). This allele is, as expected, unfortunately also lethal. We attempted to overcome lethality by generating MARCM (mosaic clones in neurons, but as expected, because *Mmp2* is largely secreted, there was no pruning

defect phenotype (Fig. S5B-C). A similar problem would also be relevant to glial MARCM clones (as the non-clonal glial population would still secrete the MMPs).

Additionally, available Mmp1 mutants are, sadly, also homozygous lethal. That said, in our revised manuscript we now include data demonstrating that expression of a dominant negative variant of Mmp1 inhibits pruning (Fig. 3J-K). We strengthened the evidence regarding the reliability of Mmp1 RNAi using an antibody mix (Fig. S4), and for Mmp2 – we refer to a manuscript that tested its efficiency (Harmansa et al., 2023). Lastly, we added new data using an additional RNAi line targeting Mmp2 from the VDRC collection (Fig. 3L).

Can the phenotype be rescued?

Unfortunately, without a viable mutant LOF phenotype, a rescue experiment is impossible. Regardless, in an attempt to rescue the RNAi phenotype, we designed and generated an RNAi-resistant Mmp2 overexpression transgene. Unfortunately, due to the destruction of our lab – several days after we received this transgenic line from Bestgene – this experiment is not included in the revision.

Does TIMP expression lead to similar phenotypes?

This is an interesting question which we addressed in our experiments but did not include in the text. Unfortunately, overexpression of TIMP did not have any effect on MB development. We now added this data as Figure S7.

What is the temporal requirement for Mmp1/2?

This is an excellent suggestion, not an easy experiment, but one that we initiated, using a temperature sensitive Gal80 to control the expression of the RNAi only during metamorphosis. However, to the unfortunate destruction of our lab, this experiment was never completed.

What are the target proteins of Mmp2?

This is the million-dollar question – but unfortunately is beyond the scope of this short report.

Is Mmp2 still required when astrocyte motility is blocked? What is the morphology of glia after Mmp1/2 knockdown?

Thank you for this wonderful suggestion. We initiated two types of experiments using sparse labeling techniques (both MARCM and SPARC) to identify the morphology of single astrocytes in WT vs. MMP KD. However, these are complicated crosses that were not completed prior to the destruction of our lab.

Reviewer #2 (Significance (Required)):

The strength of the study is to identify a pruning phenotype after RNAi-based knockdown. The limitations is that this study is very superficial, it is the beginning of a paper. The initial claim to use *Drosophila* because to its advanced genetics is not met. The results section is shorter than the discussion.

While we agree with much of the reviewer's statement this also relates to our general comment about "preliminary" type 1 and type 2 – True, this could be the beginning of a big paper and it would definitely be a more comprehensive and deep story. Most of the papers from my lab are indeed a 5 year endeavor. However, this short report (which is now longer, more detailed, and includes additional experiments) is a result of the work of an outstanding master's student who came up with the idea for the project entirely by herself. Thus – given the data that she has

acquired, and the fact that my lab will not continue to study MMPs or schizophrenia, the question needs to be whether the data supports the claims and whether this is an advance of science worthwhile of publication in a respectable journal. Our clear and decisive opinion is that the answer to that question is yes.

Reviewer #3 (Evidence, reproducibility and clarity (Required)):

In this work, Schuldiner and colleagues explore the role of Mmp1 and Mmp2 in neuronal remodeling in the mushroom body of *Drosophila*. Overall, this work is very interesting, but in its current form seems quite preliminary. The biggest limitation of the study is that single RNAi lines are used with no validation that the lines are working, despite the fact that Mmp antibodies are available as are endogenously tagged Mmp lines that could have been used to validate the genetic manipulations. Specific concerns are listed below.

We thank reviewer 3 for his generally positive assessment of our work and we now performed additional experiments to strengthen and validate the original RNAi findings – for specifics see our reply to the points below.

Major concerns

1) The scoring system for pruning of mushroom body neurons seems very variable, even in controls (where scoring can range from very mild to moderate), and it is very hard to assess from the images what one is looking at (rather than using our own judgment, we rely on the authors' words). It would be necessary to have better labeling and examples of what phenotypes are considered "mild", "severe", "wild type-like". It would also help to understand how phenotype assessment is guided by the overlap between the signals from TdTomato fluorescence and FasII stain.

We thank the reviewer for raising this point, that has also been highlighted by other reviewers in some form. First, we have generated Figure S3A-E to show examples of the ranking, which was now performed by two independent investigators. Second, ranking was performed by looking at TdTomato positive vertical axons that are outside of the α lobe (high FasII) – this is now better explained in the materials and methods. Additionally, while we would love to have a better scoring, and automatic, system – and even published a semi-automated scoring algorithm in Alyagor et al. 2018 (Figure 3O in the Alyagor paper), because the driver also labels vertical axons (α/β) and because unpruned γ axons often express FasII, this quantification method does not always work. What we have done in previous cases, as we have also done here, is to provide independent ranking by two investigators and compare their ranking (Fig. S3F-G). Finally, we are working with our AI hub to develop automatic scoring systems that will not require human ranking – however this is beyond the scope for this manuscript.

2) The biggest limitations of the approach are that single RNAi lines are used to screen, with no accompanying validation of the tool (see above)

We agree. Unfortunately not all RNAis are “equal” and thus not all of them work. To support the RNAi data, we have better clarified previous experiments that demonstrate the importance of neuronal Mmp2 via tissue specific (ts) CRISPR (Meltzer, et al, 2019). Unfortunately, the Mmp2 null mutant that is available is lethal during pupal development (Page-MaCaw et al, 2003). We therefore independently generated an Mmp2 germline mutant using CRISPR (harboring an indel

resulting in a premature stop codon and predicted to encode a truncated, 77 amino-acid long protein), now described in Fig. S5A (and in the Materials and Methods). This allele is, as expected, unfortunately also lethal. We attempted to overcome lethality by generating MARCM (mosaic) clones in neurons, but as expected, because Mmp2 is largely secreted, there was no pruning defect phenotype (Fig. S5B-C). A similar problem would also be relevant to glial MARCM clones (as the non-clonal glial population would still secrete the MMPs).

Additionally, available Mmp1 mutants are, sadly, also homozygous lethal. That said, in our revised manuscript we now include data demonstrating that expression of a dominant negative variant of Mmp1 inhibits pruning (Fig. 3J-K). We strengthened the evidence regarding the reliability of Mmp1 RNAi using an antibody mix (Fig. S4), and for Mmp2 – we refer to a manuscript that tested its efficiency (Harmansa et al., 2023). Lastly, we added new data using an additional RNAi line targeting Mmp2 from the VDRC collection (Fig. 3L).

3) RNAi-based knockdown is used to infer epistatic information-this is not appropriate as epistasis experiments need to be done with null alleles to make firm conclusions. Additional concerns:

- Even with the same driver, knockdown efficiency for 2 different genes could be variable and dependent of the specific RNAi used.
- The comparison between drivers is even harder, as driver strength varies greatly.
- The knockdown efficiency drops with increasing numbers of RNAi used.
- The specific genotypes used for this experiment should be clarified, as it would be very important to ensure that the UAS dosage is equal across conditions.

We agree that RNAi is not optimal to assess epistasis. And indeed, we did not mean to claim epistasis relationship between Mmp1 and Mmp2, nor between neurons and glia. We now use better language to clarify this. To define epistatic relationships, the use of mutants would be required, unfortunately the use of nulls is not possible because they are lethal and secreted (thus not enabling mosaic analyses). We agree that increasing the number of RNAi lines is expected to reduce their efficiency – this is why it is even more significant when we see an increased defective phenotype in the double knockdown experiments. Finally, we totally agree about the genotype comment and apologize that it was erroneously omitted in the original submission– all of which have been now added (Table 2 in materials and methods).

4) To further deepen the rigor of this work, a few simple yet important things could have been done. First, it would be important to rule out that knocking down Mmps does not affect astrocyte numbers and health (could be assessed by counting numbers and observing their morphology). Also, the authors previously showed that astrocytes actively infiltrate the axon bundle prior to pruning to facilitate axon defasciculation and pruning (Marmor-Kollet et al., 2023). It would have provided an important insight to examine if astrocytes can infiltrate the axon bundle if Mmp2 and/or Mmp1 are knocked down.

Thank you for these wonderful suggestions. We embarked on a few experiments as detailed below, unfortunately these are complicated crosses that were not completed prior to the destruction of our lab. 1) We initiated two types of experiments using sparse labeling techniques (both MARCM and SPARC) to identify the morphology of single astrocytes in WT vs. MMP KD. 2) Testing astrocytic infiltrations requires three binary systems, we obtained and generated stocks required for these experiments, but these were prematurely terminated. 3) We initiated

experiments to count the number of glial nuclei in the vicinity of the degenerating axonal lobe (at the onset of pruning). Preliminary experiments with a small n (3 controls, 4 Mmp1 RNAi, and 5 Mmp2 RNAi) suggest that the number of glial nuclei is not significantly different between these conditions.

Minor

The introduction puts big emphasis on the role of glia, but then narrows down candidate genes for the screen a γ -KCs transcriptional data set is used, and the initial screen is done via knockdown of those candidates in neurons (there is a disconnect between rationale and approach).

We totally agree with this reviewer which is why we now changed the paper to include both neuronal and glial loss-of-function screens. Figure 1 is now augmented with the glial data. Rationale for looking into axon pruning and how that translates into insights about synaptic pruning defects in schizophrenia should be more clearly stated.

Indeed, our belief that synapse pruning and axon pruning share molecular mechanisms remains yet unproven. However, both are steps during neuronal remodeling, which has been previously implicated in schizophrenia. That said, we now added an additional disclaimer to acknowledge the limitation of our findings in the context of human disease and synapse elimination (lines 275-279).

Figure 1C: data visualization for this heat map should be improved. Parts of the data are faded, and the differences between gene clusters are unclear.

We apologize for the incomplete figure. We did not want to overload the figure with data, which is why we are showing only the important clusters and did not include gene names. To keep the figure simple, but at the same time provide the complete information, we now include the full data in Fig. S1 (that includes the original heatmap with all the dynamic clusters I-IX, and including all the gene names). For the full raw data, including non-dynamic clusters, the reader is referred to look in Supplemental excel file 1. We hope this provides the clarity that this reviewer rightfully asks for.

Reviewer #3 (Significance (Required)):

In this work, Schuldiner and colleagues explore the role of Mmp1 and Mmp2 in neuronal remodeling in the mushroom body of *Drosophila*. Overall, this work is very interesting, but in its current form seems quite preliminary. The biggest limitation of the study is that single RNAi lines are used with no validation that the lines are working, despite the fact that Mmp antibodies are available as are endogenously tagged Mmp lines that could have been used to validate the genetic manipulations.

September 16, 2025

RE: Life Science Alliance Manuscript #LSA-2025-03504-TR

Prof. Oren Schuldiner
Weizmann Institute of Science
Molecular Cell Biology
Herzl 234
Rehovot, NA 7610001
Israel

Dear Dr. Schuldiner,

Thank you for submitting your revised manuscript entitled "A Drosophila screen of schizophrenia-related genes highlights MMP requirement in neuronal remodeling". As discussed previously, we would be happy to publish your paper in Life Science Alliance pending final revisions necessary to meet our formatting guidelines.

- Please add ORCID ID for corresponding (and secondary corresponding) author--you should have received instructions on how to do so.
- The titles in both the system and the manuscript file must be consistent with each other. We suggest using the title in the manuscript file.
- Please use the [10 author names, et al.] format in your references (i.e., limit the author names to the first 10).
- Please check the labels in Figure S4 are correct.
- Please add callouts for Figures 4J; 5A-O; S2A-B; S4A-D; S5A-C; S6A; S7A-D; S8A-F; S9A-C and S10A-F to your main manuscript text.
- Please check if any of the images from control flies in Fig 2, Fig S2, Fig S6, and Fig S7 are duplicated among these figures. If so, please ensure this duplication is noted in at least one of the respective figure legends.

LSA now encourages authors to provide a 30-60 second video where the study is briefly explained. We will use these videos on social media to promote the published paper and the presenting author (for examples, see <https://docs.google.com/document/d/1-UWCfbE4pGcDdcgzcmiuJl2XMBJnxKYeqRvLLrLS08s/edit?usp=sharing>). Corresponding or first-authors are welcome to submit the video. Please submit only one video per manuscript. The video can be emailed to contact@life-science-alliance.org

A. FINAL FILES:

B. MANUSCRIPT ORGANIZATION AND FORMATTING:

Thank you for your attention to these final processing requirements. Please revise and format the manuscript and upload materials as soon as you are able.

Sincerely,

September 17, 2025

RE: Life Science Alliance Manuscript #LSA-2025-03504-TRR

Prof. Oren Schuldiner
Weizmann Institute of Science
Molecular Cell Biology
Herzl 234
Rehovot, NA 7610001
Israel

Dear Dr. Schuldiner,

Thank you for submitting your Research Article entitled "A Drosophila screen of schizophrenia genes highlights neural and glial MMPs in neuronal remodeling". It is a pleasure to let you know that your manuscript is now accepted for publication in Life Science Alliance. Thanks especially for your very quick attendance to the minor formatting requests we made. Congratulations on this interesting work!

DISTRIBUTION OF MATERIALS:

Again, congratulations on a very nice paper. I hope you found the review process to be constructive and are pleased with how the manuscript was handled editorially. We look forward to future exciting submissions from your lab.

Sincerely,
